# OpenMixup: Open Mixup Toolbox and Benchmark for Visual Representation Learning

Siyuan Li[1*]  Zedong Wang[1*]  Zicheng Liu[1]  Di Wu[1]  Cheng Tan[1]  Weiyang Jin[1]  Stan Z. Li[1†]

[1]AI Lab, Research Center for Industries of the Future, Westlake University, Hangzhou, China

{lisiyuan, wangzedong, liuzicheng, wudi, tancheng, weiyangjin, stan.zq.li}@westlake.edu.cn

## Abstract

Mixup augmentation has emerged as a powerful technique for improving the generalization ability of deep neural networks. However, the lack of standardized implementations and benchmarks has hindered progress, resulting in poor reproducibility, unfair comparisons, and conflicting insights. In this paper, we introduce OpenMixup, the *first* mixup augmentation benchmark for visual representation learning, where 18 representative mixup baselines are trained *from scratch* and systematically evaluated on 11 image datasets across varying scales and granularity, spanning fine-grained scenarios to complex non-iconic scenes. We also open-source a modular codebase for streamlined mixup method design, training, and evaluations, which comprises a collection of widely-used vision backbones, optimization policies, and analysis toolkits. Notably, the codebase not only underpins all our benchmarking but supports broader mixup applications beyond classification, such as self-supervised learning and regression tasks. Through extensive experiments, we present insights on performance-complexity trade-offs and identify preferred mixup strategies for different needs. To the best of our knowledge, OpenMixup has contributed to a number of studies in the mixup community. We hope this work can further advance reproducible mixup research and fair comparisons, thereby laying a solid foundation for future progress. The source code is publicly available.

## 1  Introduction

Data mixing, or mixup, has proven effective in enhancing the generalization ability of DNNs, with notable success in visual classification tasks. The pioneering Mixup [1] proposes to generate mixed training examples through the convex combination of two input samples and their corresponding one-hot labels. By encouraging models to learn smoother decision boundaries, mixup effectively reduces overfitting and thus improves the overall performance. ManifoldMix [2] and PatchUp [3] extend this operation to the hidden space. CutMix [4] presents an alternative approach, where an input rectangular region is randomly cut and pasted onto the target in the identical location. Subsequent works [5, 6, 7] have focused on designing more complex *hand-crafted* policies to generate diverse and informative mixed samples, which can all be categorized as *static* mixing methods.

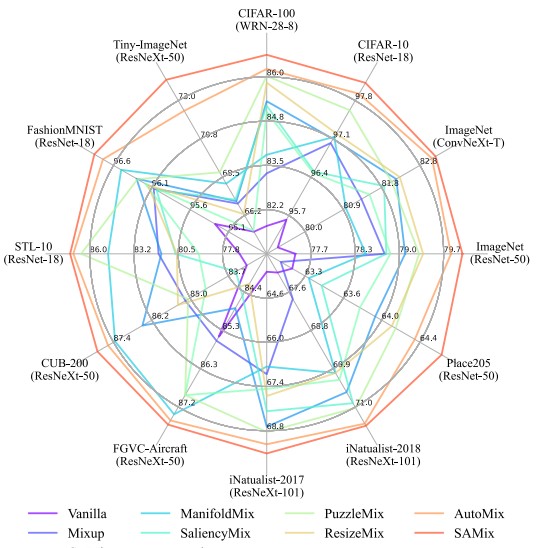

Figure 1: Radar plot of top-1 accuracy for representative mixup baselines on 11 classification datasets.

---

*Equal contribution.  †Corrsponding author.

Submitted to the 38th Conference on Neural Information Processing Systems (NeurIPS 2024) Track on Datasets and Benchmarks. Do not distribute.

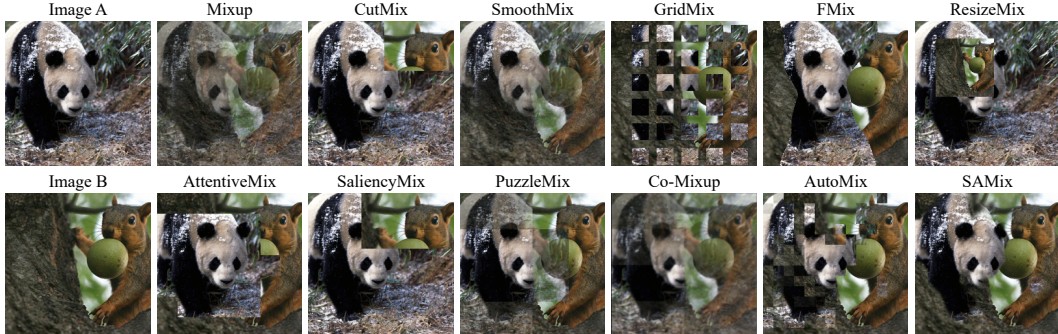

Figure 2: Visualization of mixed samples from representative *static* and *dynamic* mixup augmentation methods on ImageNet-1K. We employ a mixing ratio of $\lambda = 0.5$ for a comprehensive comparison. Note that mixed samples are more precisely in *dynamic* mixing policies than these *static* ones.

Despite efforts to incorporate saliency information into *static* mixing framework [8, 9, 10], they still struggle to ensure the inclusion of desired targets in the mixed samples, which may result in the issue of label mismatches. To address this problem, a new class of optimization-based methods, termed *dynamic* mixing, has been proposed, as illustrated in the second row of Figure 2. PuzzleMix [11] and Co-Mixup [12] are two notable studies that leverage optimal transport to improve offline mask determination. More recently, TransMix [13], TokenMix [14], MixPro [15], and SMMix [16] are specifically tailored for Vision Transformers [17]. The AutoMix series [18, 19] introduces a brand-new mixup learning paradigm, where mixed samples are computed by an online-optimizable generator in an end-to-end manner. These emerging *dynamic* approaches represent a promising avenue for generating semantically richer training samples that align with the underlying structure of input data.

**Why do we call for a mixup augmentation benchmark?** While *dynamic* methods have shown signs of surpassing the *static* ones, their indirect optimization process incurs significant computational overhead, which limits their efficiency and applicability. Therefore, without a systematic understanding, it is uncertain if *dynamic* mixup serves as the superior alternative in vision tasks. Moreover, a thorough and standardized evaluation of different *dynamic* methods is also missing in the community. Benchmark is exactly the way to establish such an understanding, which plays a pivotal role in driving research progress by integrating an agreed-upon set of tasks, impartial comparisons, and assessment criteria. To the best of our knowledge, however, there have been no such comprehensive benchmarks for mixup augmentation to facilitate unbiased comparisons and practical use in visual recognition.

**Why do we need an open-source mixup codebase?** Notably, most existing mixup techniques are crafted with diverse settings, tricks, and implementations, each with its own coding style. This lack of standardization not only hinders user-friendly reproduction and deployment but impedes further development, thus imposing costly trial-and-error on practitioners to determine the most appropriate mixup strategy for their specific needs in real-world applications. Hence, it is essential to develop a unified mixup visual representation learning codebase for standardized data pre-processing, mixup development, network architecture selection, model training, evaluation, and empirical analysis.

In this paper, we present OpenMixup, the *first* comprehensive benchmark for mixup augmentation. Unlike previous work [20, 21], we train and evaluate 18 approaches that represent the foremost strands on 11 diverse classification datasets, as illustrated in Figure 1. We also open-source a standardized codebase for mixup-based visual representation learning. The overall framework is built up with modular components for data pre-processing, mixup augmentation, network backbone selection, optimization, and evaluations, which not only powers our benchmarking study but has supported broader relatively under-explored mixup applications beyond classification, such as semi-supervised learning [22, 23], self-supervised learning [24, 25], and visual attribute regression [26, 27].

Furthermore, insightful observations are obtained by incorporating multiple evaluation metrics and analysis toolkits with our OpenMixup, including GPU memory usage (as Figure 4), loss landscape (as Figure 6), analysis of robustness and calibration (as Table A8). For example, despite the key role *static* mixing plays in today's deep learning systems, we surprisingly find that its generalizability over diverse datasets and backbones is significantly inferior to that of *dynamic* algorithms. By ranking the performance and efficiency trade-offs, we reveal that several recent *dynamic* methods have already outperformed the *static* ones. This may suggest a promising breakthrough for mixup augmentation, provided that the *dynamic* computational overhead can be further reduced. Overall, we believe these observations can facilitate meaningful evaluation and comparisons of mixup variants, enabling a systematic understanding and paving the way for future advancements in the community.

It is worth emphasizing that such a first-of-its benchmark can be rather time- and resource-consuming. Since most existing studies have focused on visual classification tasks, we centralize the benchmarking scope on this field while extending it to broader mixup applications beyond classification with transfer learning. Meanwhile, we have already supported these downstream tasks and datasets in our proposed codebase, allowing users to customize their mixup algorithms, models, and training setups in these relatively under-explored scenarios. Our key contributions can thus be summarized as follows:

- We introduce OpenMixup, the *first* comprehensive benchmarking study for mixup augmentation, where 18 representative baselines are trained from scratch and rigorously evaluated on 11 visual classification datasets, ranging from non-iconic scenes to gray-scale, fine-grained, and long tail scenarios. By providing a standard testbed and a rich set of evaluation protocols, OpenMixup enables objective assessment and fair comparisons of different mixup methods.

- To support reproducible research and user-friendly development, we open-source a unified codebase for mixup-based visual representation learning. The codebase incorporates standardized modules for data pre-processing, mixup augmentation, backbone selection, optimization policies, and distributed training functionalities. Beyond the benchmark itself, our codebase is readily extensible and has supported semi- and self-supervised learning and visual attribute regression tasks, which further enhances its versatility and potential benefits.

- Observations and insights are obtained through extensive analysis. We investigate the generalization ability of all evaluated mixup baselines across diverse datasets and backbones, compare their GPU memory footprint and computational cost, visualize the loss landscape to understand optimization behavior, and evaluate robustness against input corruptions and calibration performance. Furthermore, we establish comprehensive rankings in terms of their performance and applicability (efficiency and versatility), offering clear method guidelines for specific requirements. These findings not only present a firm grasp of the current mixup landscape but shed light on promising avenues for systematic advancements in the future.

## 2 Background and Related Work

### 2.1 Problem Definition

**Mixup training.** We first consider the general image classification tasks with $k$ different classes: given a finite set of $n$ image samples $X = [x_i]_{i=1}^n \in \mathbb{R}^{n \times W \times H \times C}$ and their corresponding ground-truth class labels $Y = [y_i]_{i=1}^n \in \mathbb{R}^{n \times k}$, encoded by a one-hot vector $y_i \in \mathbb{R}^k$. We attempt to seek the mapping from input data $x_i$ to its class label $y_i$ modeled through a deep neural network $f_\theta : x \longmapsto y$ with parameters $\theta$ by optimizing a classification loss $\ell(.)$, say the cross entropy (CE) loss,

$$\ell_{CE}(f_\theta(x), y) = -y \log f_\theta(x). \tag{1}$$

Then we consider the mixup classification task: given a sample mixing function $h$, a label mixing function $g$, and a mixing ratio $\lambda$ sampled from $Beta(\alpha, \alpha)$ distribution, we can generate the mixed data $X_{mix}$ with $x_{mix} = h(x_i, x_j, \lambda)$ and the mixed label $Y_{mix}$ with $y_{mix} = g(y_i, y_j, \lambda)$, where $\alpha$ is a hyper-parameter. Similarly, we learn $f_\theta : x_{mix} \longmapsto y_{mix}$ by the mixup cross-entropy (MCE) loss,

$$\ell_{MCE} = \lambda \ell_{CE}(f_\theta(x_{mix}), y_i) + (1 - \lambda)\ell_{CE}(f_\theta(x_{mix}), y_j). \tag{2}$$

**Mixup reformulation.** Comparing Eq. (1) and Eq. (2), the mixup training has the following features: (1) extra mixup policies, $g$ and $h$, are required to generate $X_{mix}$ and $Y_{mix}$. (2) the classification performance of $f_\theta$ depends on the generation policy of mixup. Naturally, we can split the mixup task into two complementary sub-tasks: (i) mixed sample generation and (ii) mixup classification (learning objective). Notice that the sub-task (i) is subordinate to (ii) because the final goal is to obtain a stronger classifier. Therefore, from this perspective, we regard the mixup generation as an auxiliary task for the classification task. Since $g$ is generally designed as a linear interpolation, i.e., $g(y_i, y_j, \lambda) = \lambda y_i + (1 - \lambda)y_j$, $h$ becomes the key function to determine the performance of the model. Generalizing previous offline methods, we define a parametric mixup policy $h_\phi$ as the sub-task with another set of parameters $\phi$. The final goal is to optimize $\ell_{MCE}$ given $\theta$ and $\phi$ as:

$$\min_{\theta, \, \phi} \ell_{MCE}\Big(f_\theta\big(h_\phi(x_i, x_j, \lambda)\big), g(y_i, y_j, \lambda)\Big). \tag{3}$$

### 2.2 Sample Mixing

Within the realm of visual classification, prior research has primarily concentrated on refining the sample mixing strategies rather than the label mixing ones. In this context, most sample mixing methods are categorized into two groups: *static* policies and *dynamic* policies, as presented in Table 1.

Table 1: Information of all supported vision Mixup augmentation methods in OpenMixup. Note that Mixup and CutMix in the label mixing policies indicate mixing labels of two samples by linear interpolation or calculating the cut squares. The *Perf.*, *App.*, and *Overall* headings below denote the performance, applicability (efficiency & versatility), and overall rankings of all the mixup baselines.

| Method | Category | Publication | Sample Mixing | Label Mixing | Extra Cost | ViT only | Perf. | App. | Overall |
|---|---|---|---|---|---|---|---|---|---|
| Mixup [1] | Static | ICLR'2018 | Hand-crafted Interpolation | Mixup | ✗ | ✗ | 15 | 1 | 10 |
| CutMix [4] | Static | ICCV'2019 | Hand-crafted Cutting | CutMix | ✗ | ✗ | 13 | 1 | 8 |
| DeiT (CutMix+Mixup) [28] | Static | ICML'2021 | CutMix+Mixup | CutMix+Mixup | ✗ | ✗ | 7 | 1 | 3 |
| SmoothMix [6] | Static | CVPRW'2020 | Hand-crafted Cutting | CutMix | ✗ | ✗ | 18 | 1 | 13 |
| GridMix [7] | Static | PR'2021 | Hand-crafted Cutting | CutMix | ✗ | ✗ | 17 | 1 | 12 |
| ResizeMix [10] | Static | CVMJ'2023 | Hand-crafted Cutting | CutMix | ✗ | ✗ | 10 | 1 | 5 |
| ManifoldMix [2] | Static | ICML'2019 | Latent-space Mixup | Mixup | ✗ | ✗ | 14 | 1 | 9 |
| FMix [5] | Static | arXiv'2020 | Fourier-guided Cutting | CutMix | ✗ | ✗ | 16 | 1 | 11 |
| AttentiveMix [8] | Static | ICASSP'2020 | Pretraining-guided Cutting | CutMix | ✓ | ✗ | 9 | 3 | 6 |
| SaliencyMix [9] | Static | ICLR'2021 | Saliency-guided Cutting | CutMix | ✗ | ✗ | 11 | 1 | 6 |
| PuzzleMix [11] | Dynamic | ICML'2020 | Optimal-transported Cutting | CutMix | ✓ | ✗ | 8 | 4 | 6 |
| AlignMix [29] | Dynamic | CVPR'2022 | Optimal-transported Interpolation | CutMix | ✓ | ✗ | 12 | 2 | 8 |
| AutoMix [18] | Dynamic | ECCV'2022 | End-to-end-learned Cutting | CutMix | ✓ | ✗ | 3 | 6 | 4 |
| SAMix [30] | Dynamic | arXiv'2021 | End-to-end-learned Cutting | CutMix | ✓ | ✗ | 1 | 5 | 1 |
| AdAutoMix [19] | Dynamic | ICLR'2024 | End-to-end-learned Cutting | CutMix | ✓ | ✗ | 2 | 7 | 4 |
| TransMix [13] | Dynamic | CVPR'2022 | CutMix+Mixup | Attention-guided | ✗ | ✓ | 5 | 8 | 7 |
| SMMix [16] | Dynamic | ICCV'2023 | CutMix+Mixup | Attention-guided | ✗ | ✓ | 4 | 8 | 6 |
| DecoupledMix [23] | Static | NIPS'2023 | Any Sample Mixing Policies | DecoupledMix | ✗ | ✗ | 6 | 1 | 2 |

**Static Policies.**    The sample mixing procedure in all *static* policies is conducted in a *hand-crafted* manner. Mixup [1] first generates artificially mixed data through the convex combination of two randomly selected input samples and their associated one-hot labels. ManifoldMix variants [2, 3] extend the same technique to latent feature space for better sample mixing performance. Subsequently, CutMix [4] involves the random replacement of a certain rectangular region inside input sample while concurrently employing Dropout throughout the mixing process. Inspired by CutMix, several researchers in the community have explored the use of saliency information [9] to pilot mixing patches, while others have developed more complex *hand-crafted* sample mixing strategies [5, 7, 6].

**Dynamic Policies.**    In contrast to *static* mixing, *dynamic* strategies are proposed to incorporate sample mixing into an adaptive optimization-based framework. PuzzleMix variants [11, 12] introduce combinatorial optimization-based mixing policies in accordance with saliency maximization. Super-Mix variants [31, 8] utilize pre-trained teacher models to compute smooth and optimized samples. Distinctively, AutoMix variants [18, 30] reformulate the overall sample mixing framework into an online-optimizable fashion that learns to generate the mixed samples in an end-to-end manner.

## 2.3 Label Mixing

Mixup [1] and CutMix [4] are two widely-recognized label mixing techniques, both of which are *static*. Recently, there has been a notable emphasis among researchers on advancing label mixing approaches, which attains more favorable performance upon certain sample mixing policies. Based on Transformers, TransMix variants [13, 14, 32, 16] are proposed to utilize class tokens and attention maps to adjust the mixing ratio. A decoupled mixup objective [23] is introduced to force models to focus on those hard mixed samples, which can be plugged into different sample mixing policies. Holistically, most existing studies strive for advanced sample mixing designs rather than label mixing.

## 2.4 Other Applications

Recently, mixup augmentation also has shown promise in more vision applications, such as semi-supervised learning [22, 23], self-supervised pre-training [24, 25], and visual attribute regression [26, 27]. Although these fields are not as extensively studied as classification, our OpenMixup codebase has been designed to support them by including the necessary task settings and datasets. Its modular and extensible architecture allows researchers and practitioners in the community to effortlessly adapt and extend their models to accommodate the specific requirements of these tasks, enabling them to quickly set up experiments without building the entire pipeline from scratch. Moreover, our codebase will be well-positioned to accelerate the development of future benchmarks, ultimately contributing to the advancement of mixup augmentation across a diversity of visual representation learning tasks.

# 3   OpenMixup

This section introduces our OpenMixup codebase framework and benchmark from four key aspects: supported methods and tasks, evaluation metrics, and experimental pipeline. OpenMixup provides a unified framework implemented in PyTorch [33] for mixup model design, training, and evaluation.

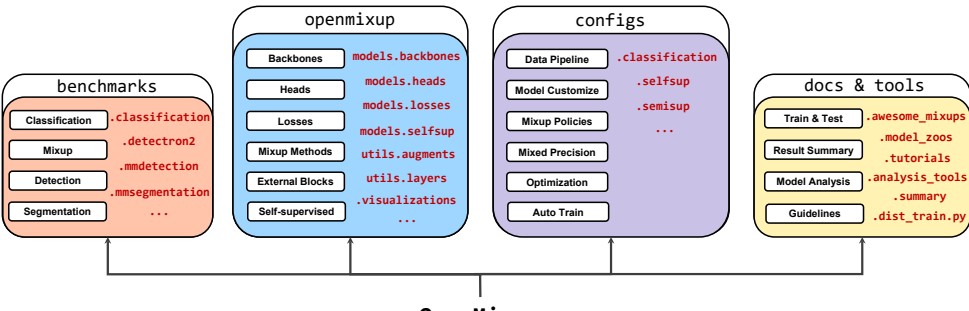

Figure 3: Overview of codebase framework of OpenMixup benchmark. (1) `benchmarks` provide the benchmarking results and corresponding config files for mixup classification and transfer learning. (2) `openmixup` contains the source codes of all supported methods. (3) `configs` is responsible for customizing setups of different mixup methods, networks, datasets, and training pipelines. (4) `docs & tools` contains paper lists of popular mixup methods, user documentation, and practical tools.

The framework references MMClassification [34] and follows the OpenMMLab coding style. We start with an overview of its composition. As shown in Figure 3, the whole training process here is fragmented into multiple components, including model architecture (`.openmixup.models`), data pre-processing (`.openmixup.datasets`), mixup policies (`.openmixup.models.utils.augments`), script tools (`.tools`) *etc.* For instance, vision models are summarized into modular building blocks (*e.g.*, backbone, neck, head *etc.*) in `.openmixup.models`. This modular architecture enables practitioners to easily craft models by incorporating different components through configuration files in `.configs`. As such, users can readily customize their specified vision models and training strategies. In addition, benchmarking configuration (`.benchmarks`) and results (`.tools.model_zoos`) are also provided in the codebase. Additional benchmarking details are discussed below.

## 3.1 Benchmarked Methods

OpenMixup has implemented 17 representative mixup augmentation algorithms and 19 convolutional neural network and Transformer model architectures (gathered in `.openmixup.models`) across 12 diverse image datasets for supervised visual classification. We summarize these mixup methods in Table 1, along with their corresponding conference/journal, the types of employed sample and label mixing policies, properties, and rankings. For sample mixing, Mixup [1] and Manifold-Mix [2] perform *hand-crafted* convex interpolation. CutMix [4], SmoothMix [6], GridMix [7] and ResizeMix [10] implement *hand-crafted* cutting policy. FMix [5] utilizes Fourier-guided cutting. AttentiveMix [8] and SaliencyMix [9] apply pretraining-guided and saliency-guided cutting, respectively. Some *dynamic* approaches like PuzzleMix [11] and AlignMix [29] utilize optimal transport-based cutting and interpolation. AutoMix [18] and SAMix [30] perform end-to-end online-optimizable cutting-based approaches. As for the label mixing, most methods apply Mixup [1] or CutMix [4], while the latest mixup methods for visual transformers (TransMix [13], TokenMix [14], and SMMix [16]), as well as DecoupledMix [23] exploit attention maps and a decoupled framework respectfully instead, which incorporate CutMix variants as its sample mixing strategy. Such a wide scope of supported methods enables a comprehensive benchmarking analysis on visual classification.

## 3.2 Benchmarking Tasks

We provide detailed descriptions of the 12 open-source datasets as shown in Table 2. These datasets can be classified into four categories below: **(1) Small-scale classification**: We conduct benchmarking studies on small-scale datasets to provide an accessible benchmarking reference. CIFAR-10/100 [35] consists of 60,000 color images in 32×32 resolutions. Tiny-ImageNet (Tiny) [36] and STL-10 [37] are two re-scale versions of ImageNet-1K in the size of 64×64 and 96×96. FashionMNIST [38] is the advanced version of MNIST, which contains gray-scale images of clothing. **(2) Large-scale classification**: The large-scale dataset is employed to evaluate mixup algorithms against the most standardized procedure, which can also support the prevailing ViT architecture. ImageNet-1K (IN-1K) [39] is a well-known challenging dataset for image classification with 1000 classes. **(3) Fine-grained classification**: To investigate the effectiveness of mixup methods in complex inter-class relationships and long-tail scenarios, we conduct a comprehensive evaluation of fine-grained classification datasets, which can also be classified into small-scale and large-scale scenarios. (i) *Small-scale scenarios*: The datasets for small-scale fine-grained evaluation scenario are CUB-200-2011 (CUB) [40] and FGVC-Aircraft (Aircraft) [41], which contains a total of 200 wild

Table 2: The detailed information of supported visual classification datasets in OpenMixup.

| Datasets | Category | Source | Classes | Resolution | Train images | Test images |
|---|---|---|---|---|---|---|
| CIFAR-10 [35] | Iconic | link | 10 | 32×32 | 50,000 | 10,000 |
| CIFAR-100 [35] | Iconic | link | 100 | 32×32 | 50,000 | 10,000 |
| FashionMNIST [38] | Gray-scale | link | 10 | 28×28 | 50,000 | 10,000 |
| STL-10 [37] | Iconic | link | 10 | 96×96 | 50,00 | 8,000 |
| Tiny-ImageNet [36] | Iconic | link | 200 | 64×64 | 10,000 | 10,000 |
| ImageNet-1K [39] | Iconic | link | 1000 | 469×387 | 1,281,167 | 50,000 |
| CUB-200-2011 [40] | Fine-grained | link | 200 | 224×224 | 5,994 | 5,794 |
| FGVC-Aircraft [41] | Fine-grained | link | 100 | 224×224 | 6,667 | 3,333 |
| iNaturalist2017 [42] | Fine-grained & longtail | link | 5089 | 224×224 | 579,184 | 95,986 |
| iNaturalist2018 [42] | Fine-grained & longtail | link | 8142 | 224×224 | 437,512 | 24,426 |
| Places205 [43] | Scenic | link | 205 | 224×224 | 2,448,873 | 41,000 |

bird species and 100 classes of airplanes. (ii) *Large-scale scenarios*: The datasets for large-scale fine-grained evaluation scenarios are iNaturalist2017 (iNat2017) [42] and iNaturalist2018 (iNat2018) [42], which contain 5,089 and 8,142 natural categories. Both the iNat2017 and iNat2018 own 7 major categories and are also long-tail datasets with scenic images (*i.e.*, the fore-ground target is in large backgrounds). **(4) Scenic classification**: Scenic classification evaluations are also conducted to investigate the mixup augmentation performance in complex non-iconic scenarios on Places205 [43].

## 3.3   Evaluation Metrics and Tools

We comprehensively evaluate the beneficial properties of mixup augmentation algorithms on the aforementioned vision tasks through the use of various metrics and visualization analysis tools in a rigorous manner. Overall, the evaluation methodologies can be classified into two distinct divisions, namely performance metric and empirical analysis. For the performance metrics, classification accuracy and robustness against corruption are two performance indicators examined. As for empirical analysis, experiments on calibrations, CAM visualization, loss landscape, the plotting of training loss, and validation accuracy curves are conducted. The utilization of these approaches is contingent upon their distinct properties, enabling user-friendly deployment for designated purposes and demands.

**Performance Metric.**   **(1) Accuracy and training costs**: We adopt top-1 accuracy, total training hours, and GPU memory to evaluate all mixup methods' classification performance and training costs. **(2) Robustness**: We evaluate the robustness against corruptions of the methods on CIFAR-100-C and ImageNet-C [39], which is designed for evaluating the corruption robustness and provides 19 different corruptions, *e.g.*, noise and blur *etc.* **(3) Transferability to downstream tasks**: We evaluate the transferability of existing methods to object detection based on Faster R-CNN [44] and Mask R-CNN [45] on COCO *train2017* [46], initializing with trained models on ImageNet. We also transfer these methods to semantic segmentation on ADE20K [47]. Please refer to Appendix B.4 for details.

**Empirical Analysis.**   **(1) Calibrations**: To verify the calibration of existing methods, we evaluate them by the expected calibration error (ECE) on CIFAR-100 [35], *i.e.*, the absolute discrepancy between accuracy and confidence. **(2) CAM visualization**: We utilize mixed sample visualization, a series of CAM variants [48, 49] (*e.g.*, Grad-CAM [50]) to directly analyze the classification accuracy and especially the localization capabilities of mixup augmentation algorithms through top-1 top-2 accuracy predicted targets. **(3) Loss landscape**: We apply loss landscape evaluation [51] to further analyze the degree of loss smoothness of different mixup augmentation methods. **(4) Training loss and accuracy curve**: We plot the training losses and validation accuracy curves of various mixup methods to analyze the training stability, the ability to prevent over-fitting, and convergence speed.

## 3.4   Experimental Pipeline of OpenMixup Codebase

With a unified training pipeline in OpenMixup, a comparable workflow is shared by different classification tasks, as illustrated in Figure A1. Here, we take classification tasks as an instance to illustrate the whole training procedure. Firstly, users should go through the supported data pipeline and select the dataset and pre-processing techniques. Secondly, `openmixup.models` serves as a model architecture component for building desired methods. Thirdly, it is undemanding to designate the supported datasets, mixup augmentation strategies, model architectures, and optimization schedules under `.configs.classification` with Python configuration files to customize a desired setting. Afterward, `.tools` provides hardware support distributed training to execute the confirmed training process in configs. Apart from that, there are also various utility functionalities given in `.tools` (*e.g.*, feature visualization, model analysis, result summarization). We also provide online user documents

Table 3: Top-1 accuracy (%) on CIFAR-10/100 and Tiny-ImageNet (Tiny) based on ResNet (R), Wide-ResNet (WRN), and ResNeXt (RX) backbones.

| Datasets | CIFAR-10 | CIFAR-100 | Tiny |
|---|---|---|---|
| Backbones | R-18 | WRN-28-8 | RX-50 |
| Epochs | 800 ep | 800 ep | 400 ep |
| Vanilla | 95.50 | 81.63 | 65.04 |
| Mixup | 96.62 | 82.82 | 66.36 |
| CutMix | 96.68 | 84.45 | 66.47 |
| ManifoldMix | 96.71 | 83.24 | 67.30 |
| SmoothMix | 96.17 | 82.09 | 68.61 |
| AttentiveMix | 96.63 | 84.34 | 67.42 |
| SaliencyMix | 96.20 | 84.35 | 66.55 |
| FMix | 96.18 | 84.21 | 65.08 |
| GridMix | 96.56 | 84.24 | 69.12 |
| ResizeMix | 96.76 | 84.87 | 65.87 |
| PuzzleMix | 97.10 | 85.02 | 67.83 |
| Co-Mixup | 97.15 | 85.05 | 68.02 |
| AlignMix | 97.05 | 84.87 | 68.74 |
| AutoMix | 97.34 | 85.18 | 70.72 |
| SAMix | 97.50 | **85.50** | 72.18 |
| AdAutoMix | **97.55** | 85.32 | **72.89** |
| Decoupled | 96.95 | 84.88 | 67.46 |

Table 4: Top-1 accuracy (%) on ImageNet-1K using PyTorch-style, RSB A2/A3, and DeiT settings based on CNN and Transformer architectures, including ResNet (R), MobileNet.V2 (Mob.V2), DeiT-S, and Swin-T.

| Backbones | R-50 | R-50 | Mob.V2 1x | DeiT-S | Swin-T |
|---|---|---|---|---|---|
| Epochs | 100 ep | 100 ep | 300 ep | 300 ep | 300 ep |
| Settings | PyTorch | RSB A3 | RSB A2 | DeiT | DeiT |
| Vanilla | 76.83 | 77.27 | 71.05 | 75.66 | 80.21 |
| Mixup | 77.12 | 77.66 | 72.78 | 77.72 | 81.01 |
| CutMix | 77.17 | 77.62 | 72.23 | 80.13 | 81.23 |
| DeiT / RSB | 77.35 | 78.08 | 72.87 | 79.80 | 81.20 |
| ManifoldMix | 77.01 | 77.78 | 72.34 | 78.03 | 81.15 |
| AttentiveMix | 77.28 | 77.46 | 70.30 | 80.32 | 81.29 |
| SaliencyMix | 77.14 | 77.93 | 72.07 | 79.88 | 81.37 |
| FMix | 77.19 | 77.76 | 72.79 | 80.45 | 81.47 |
| ResizeMix | 77.42 | 77.85 | 72.50 | 78.61 | 81.36 |
| PuzzleMix | 77.54 | 78.02 | 72.85 | 77.37 | 79.60 |
| AutoMix | 77.91 | 78.44 | 73.19 | 80.78 | 81.80 |
| SAMix | **78.06** | **78.64** | **73.42** | 80.94 | **81.87** |
| AdAutoMix | 78.04 | 78.54 | - | 80.81 | 81.75 |
| TransMix | - | - | - | 80.68 | 81.80 |
| SMMix | - | - | - | **81.10** | 81.80 |

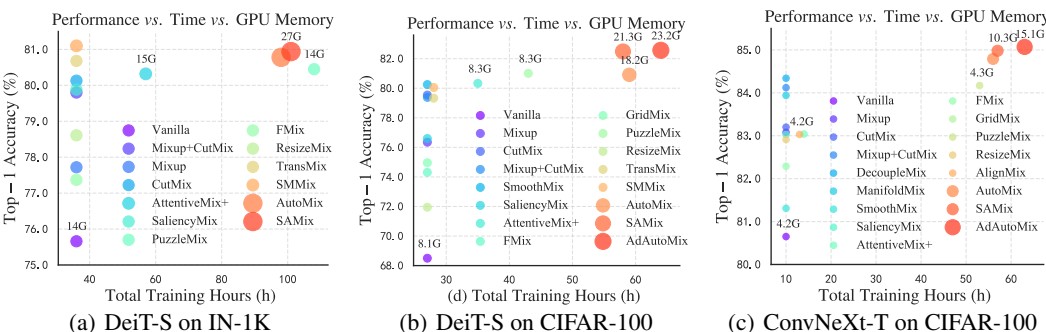

Figure 4: Trade-off evaluation with respect to accuracy performance, total training time (hours), and GPU memory (G). The results in (a) are based on DeiT-S architecture on ImageNet-1K. The results in (b) and (c) are based on DeiT-S and ConvNeXt-T backbones on CIFAR-100, respectively.

for more detailed guidelines (*e.g.*, installation and getting started instructions), benchmarking results, comprehensive awesome lists of related works, *etc*.

## 4  Experiment and Analysis

### 4.1  Implementation Details

We conduct essential benchmarking experiments of image classification on various scenarios with diverse evaluation metrics. For a fair comparison, grid search is performed for the shared hyper-parameter $\alpha \in \{0.1, 0.2, 0.5, 1, 2, 4\}$ of supported mixup variants while the rest of the hyper-parameters follow the original papers. Vanilla denotes the classification baseline without any mixup augmentations. All experiments are conducted on Ubuntu workstations with Tesla V100 or NVIDIA A100 GPUs and report the *mean* results of three trials. Appendix B provides classification results, and Appendix B.4 presents transfer learning results for object detection and semantic segmentation.

**Small-scale Benchmarks.** We first provide standard mixup image classification benchmarks on five small datasets with two settings. (a) The classical settings with the CIFAR version of ResNet variants [52, 53], *i.e.*, replacing the $7 \times 7$ convolution and MaxPooling by a $3 \times 3$ convolution. We use $32 \times 32$, $64 \times 64$, and $28 \times 28$ input resolutions for CIFAR-10/100, Tiny-ImageNet, and FashionMNIST, while using the normal ResNet for STL-10. We train models for multiple epochs from the stretch with SGD optimizer and a batch size of 100, as shown in Table 3 and Appendix B.2. (b) The modern settings following DeiT [28] on CIFAR-100, using $224 \times 224$ and $32 \times 32$ resolutions for Transformers (DeiT-S [28] and Swin-T [54]) and ConvNeXt-T [55] as shown in Table A7.

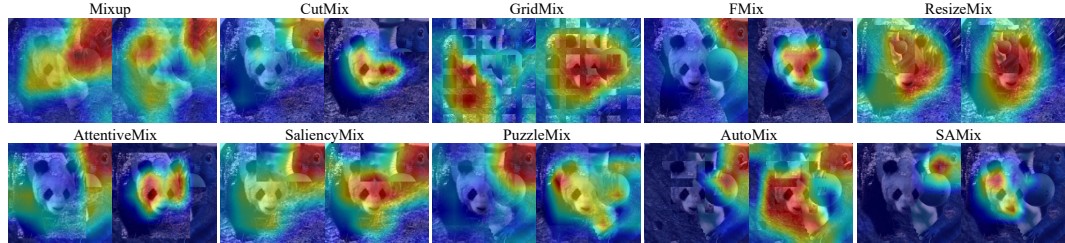

Figure 5: Visualization of class activation mapping (CAM) [50] for top-1 and top-2 predicted classes of supported mixup methods with ResNet-50 on ImageNet-1K. Comparing the first and second rows, we observe that saliency-guided or dynamic mixup approaches (*e.g.*, PuzzleMix and SAMix) localize the target regions better than the static methods (*e.g.*, Mixup and ResizeMix).

Table 5: Rankings of various mixup augmentations as take-home messages for practical usage.

|  | Mixup | CutMix | DeiT | Smooth | GridMix | ResizeMix | Manifold | FMix | Attentive | Saliency | PuzzleMix | AlignMix | AutoMix | SAMix | TransMix | SMMix |
|---|---|---|---|---|---|---|---|---|---|---|---|---|---|---|---|---|
| Performance | 13 | 11 | 5 | 16 | 15 | 8 | 12 | 14 | 7 | 9 | 6 | 10 | 2 | 1 | 4 | 3 |
| Applicability | 1 | 1 | 1 | 1 | 1 | 1 | 1 | 1 | 3 | 1 | 4 | 2 | 7 | 6 | 5 | 5 |
| Overall | 8 | 6 | 1 | 11 | 10 | 4 | 7 | 9 | 5 | 5 | 5 | 6 | 4 | 2 | 4 | 3 |

**Standard ImageNet-1K Benchmarks.** For visual augmentation and network architecture communities, ImageNet-1K is a well-known standard dataset. We support three popular training recipes: (a) PyTorch-style [52] setting for classifical CNNs; (b) timm RSB A2/A3 [56] settings; (c) DeiT [28] setting for ViT-based models. Evaluation is performed on 224×224 resolutions with `CenterCrop`. Popular network architectures are considered: ResNet [52], Wide-ResNet [57], ResNeXt [53], MobileNet.V2 [58], EfficientNet [59], DeiT [28], Swin [54], ConvNeXt [55], and MogaNet [60]. Refer to Appendix A for implementation details. In Table 4 and Table A2, we report the *mean* performance of three trials where the *median* of top-1 test accuracy in the last 10 epochs is recorded for each trial.

**Benchmarks on Fine-grained and Scenis Scenarios.** We further provide benchmarking results on three downstream classification scenarios in 224×224 resolutions with ResNet backbone architectures: (a) Transfer learning on CUB-200 and FGVC-Aircraft. (b) Fine-grained classification on iNat2017 and iNat2018. (c) Scenic classification on Places205, as illustrated in Appendix B.3 and Table A10.

## 4.2 Observations and Insights

Empirical analysis is conducted to gain insightful observations and a systematic understanding of the properties of different mixup augmentation techniques. Our key findings are summarized as follows:

**(A) Which mixup method should I choose?** Integrating benchmarking results from various perspectives, we provide practical mixup rankings (detailed in Appendix B.5) as a take-home message for real-world applications, which regards performance, applicability, and overall capacity. As shown in Table 1, as for the performance, the *online-optimizable* SAMix and AutoMix stand out as the top two choices. SMMix and TransMix follow closely behind. However, in terms of applicability that involves both the concerns of efficiency and versatility, *hand-crafted* methods significantly outperform the learning-based ones. Overall, the DeiT (Mixup+CutMix), SAMix, and SMMix are selected as the three most preferable mixup methods, each with its own emphasis.

**(B) Generalizability over datasets.** The intuitive performance radar chart presented in Figure 1, combined with the trade-off results in Fiugre 4, reveals that *dynamic* mixup methods consistently yield better performance compared to *static* ones, showcasing their impressive generalizability. However, *dynamic* approaches necessitate meticulous tuning, which incurs considerable training costs. In contrast, *static* mixup exhibits significant performance fluctuation across different datasets,

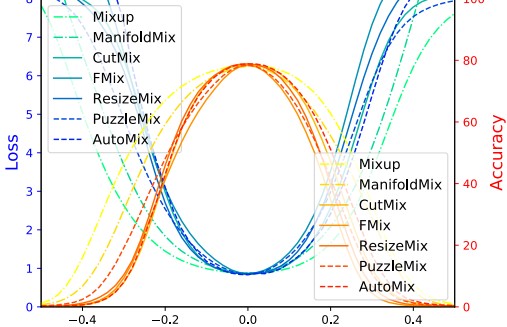

Figure 6: Visualization of 1D loss landscapes for representative mixup methods with ResNet-50 on ImageNet-1K. The validation accuracy is plotted, showing that dynamic methods achieve deeper and wider loss landscapes than static ones.

indicating poor generalizability. For instance, Mixup [1] and CutMix [4] as the *static* representatives perform even worse than the baseline on Place205 and FGVC-Aircraft, respectively.

**(C) Generalizability over backbones.** As shown in Figure 6 and Figure 4, we provide extensive evaluations on ImageNet-1K based on different types of backbones and mixup methods. As a result, *dynamic* mixup achieves better performance in general and shows more favorable generalizability against backbone selection compared to *static* methods. Noticeably, the *online-optimizable* SAMix and AutoMix exhibit impressive generalization ability over different vision backbones, which potentially reveals the superiority of their online training framework compared to the others.

**(D) Applicability.** Figure A2 shows that ViT-specific methods (*e.g.*, TransMix [13] and Token-Mix [14]) yield exceptional performance with DeiT-S and PVT-S yet exhibit intense sensitivity to different model scales (*e.g.*, with PVT-T). Moreover, they are limited to ViTs, which largely restricts their applicability. Surprisingly, *static* Mixup [1] exhibits favorable applicability with new efficient networks like MogaNet [60]. CutMix [4] fits well with popular backbones, such as modern CNNs (*e.g.*, ConvNeXt and ResNeXt) and DeiT, which increases its applicability. As in Figure 4, although AutoMix and SAMix are available in both CNNs and ViTs with consistent superiority, they have limitations in GPU memory and training time, which may limit their applicability in certain cases. This also provides a promising avenue to reduce the cost of the well-performed online learable mixup.

**(E) Robustness & Calibration.** We evaluate the robustness with accuracy on the corrupted version of CIFAR-100 and FGSM attack [61] and the prediction calibration. Table A8 shows that all mixup methods improve model robustness against corruptions. Interestingly, only four recent *dynamic* approaches exhibit better robustness compared to the baseline with FGSM attacks. We thus hypothesize that *online-optimizable* methods are well-trained to be robust against human interference, while *hand-crafted* mixup adapts to natural disruptions like corruption but is susceptible to attacks. Overall, the learning-based AutoMix and SAMix achieve the best robustness and calibration results.

**(F) Convergence & Training Stability.** As shown in Figure 6, wider bump curves indicate smoother loss landscapes, while higher warm color bump tips are associated with better convergence and performance. Evidently, *dynamic* mixup owns better training stability and convergence than *static* mixup in general. Nevertheless, the initial Mixup [1] is an exception, exhibiting better training stability than all other *dynamic* methods. We assume this arises from its straightforward convex interpolation that principally prioritizes training stability but may lead to suboptimal outcomes.

**(G) Localizability & Downstream Transferability.** It is commonly conjectured that models with better localizability can be better transferred to fine-grained prediction tasks. Thus, to gain intuitive insights, we provide tools for the class activation mapping (CAM) visualization with predicted classes on ImageNet-1K. As shown in Figure 5, SAMix and AutoMix's exceptional localizability, combined with their accuracy, generalizability, and robustness mentioned above, may indicate their superiority in detection tasks. To assess their real downstream performance and transferability, transfer learning experiments are also available on object detection [44] and semantic segmentation [62] with details in Appendix B.4. Table A11 and Table A12 suggest that AutoMix variants indeed exhibit competitive results, but ViT-specific methods perform even better, showcasing their superior transferability. This also shows the potential for improved online training mixup design.

# 5 Conclusion and Discussion

**Contributions.** This paper presents OpenMixup, the *first* comprehensive mixup augmentation benchmark, where 18 mixup baselines are trained and evaluated on 11 diverse datasets. A unified and modular codebase is also released, which not only bolsters the benchmark but can facilitate broader under-explored mixup applications. Furthermore, observations and insights are obtained through extensive analysis, contributing to a more systematic comprehension of mixups. We anticipate that our OpenMixup can further contribute to fair and reproducible research in the mixup community. We also encourage researchers and practitioners to extend their valuable feedback to us and contribute to OpenMixup for building a more constructive mixup learning codebase together through GitHub.

**Limitations and Future Works.** The benchmarking scope of this work mainly focuses on visual classification, albeit we have supported a broader range of tasks in our codebase and have conducted transfer learning experiments to object detection and semantic segmentation tasks to draw preliminary conclusions. We are aware of this and have prepared it upfront for future work. For example, our codebase can be easily extended to all the supported tasks and datasets for further benchmarking experiments and evaluations if necessary. We believe this work as the *first* mixup benchmarking study is enough to serve as a kick-start, and we plan to extend our work in these directions in the future.

## Acknowledgement

This work was supported by National Key R&D Program of China (No. 2022ZD0115100), National Natural Science Foundation of China Project (No. U21A20427), and Project (No. WU2022A009) from the Center of Synthetic Biology and Integrated Bioengineering of Westlake University. This work was done when Zedong Wang and Weiyang Jin interned at Westlake University. We thank the AI Station of Westlake University for the support of GPUs.

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
