## Supplement Material

In supplement material, we provide implementation details and more benchmark results of image classification with mixup augmentations implemented in `OpenMixup` on various datasets.

## A  Implementation Details

### A.1  Setup OpenMixup

As provided in the supplementary material or the online document, we simply introduce the installation and data preparation for OpenMixup, detailed in docs/en/latest/install.md. Assuming the PyTorch environment has already been installed, users can easily reproduce the environment with the source code by executing the following commands:

```
conda activate openmixup
pip install openmim
mim install mmcv-full
\# put the source code here
cd openmixup
python setup.py develop  \# or "pip install -e ."
```

Executing the instructions above, OpenMixup will be installed as the development mode, *i.e.*, any modifications to the local source code take effect, and can be used as a python package. Then, users can download the datasets and the released meta files and symlink them to the dataset root (`$OpenMixup/data`). The codebase is under `Apache 2.0` license.

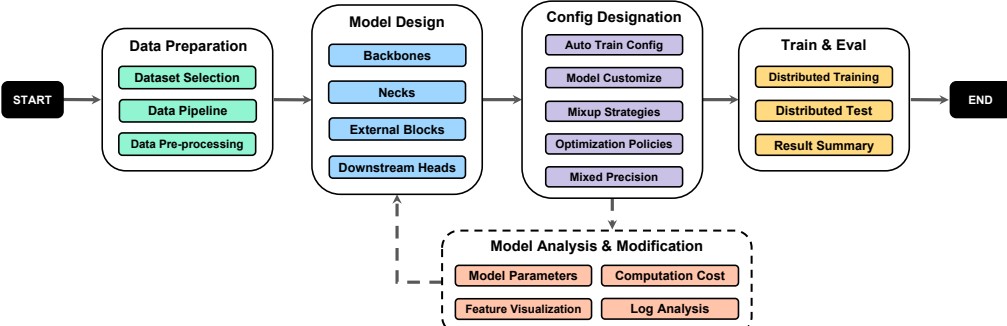

Figure A1: Overview of the experimental pipeline in OpenMixup codebase.

### A.2  Training Settings of Image Classification

**Large-scale Datasets.**  Table A1 illustrates three popular training settings on large-scaling datasets like ImageNet-1K in detail: (1) PyTorch-style [33]. (2) DeiT [28]. (3) RSB A2/A3 [56]. Notice that the step learning rate decay strategy is replaced by Cosine Scheduler [63], and `ColorJitter` as well as `PCA lighting` are removed in PyTorch-style setting for better performances. DeiT and RSB settings adopt advanced augmentation and regularization techniques for Transformers, while RSB A3 is a simplified setting for fast training on ImageNet-1K. For a fare comparison, we search the optimal hyper-parameter $\alpha$ in $Beta(\alpha, \alpha)$ from $\{0.1, 0.2, 0.5, 1, 2, 4\}$ for compared methods while the rest of the hyper-parameters follow the original papers.

**Small-scale Datasets.**  We also provide two experimental settings on small-scale datasets: (a) Following the common setups [52, 4] on small-scale datasets like CIFAR-10/100, we train 200/400/800/1200 epochs from stretch based on CIFAR version of ResNet variants [52], *i.e.*, replacing the $7 \times 7$ convolution and MaxPooling by a $3 \times 3$ convolution. As for the data augmentation, we apply `RandomFlip` and `RandomCrop` with 4 pixels padding for $32 \times 32$ resolutions. The testing image size is $32 \times 32$ (no `CenterCrop`). The basic training settings include: SGD optimizer with SGD weight decay of 0.0001, a momentum of 0.9, a batch size of 100, and a basic learning rate is 0.1 adjusted by Cosine Scheduler [63]. (b) We also provide modern training settings following DeiT [28], while using $224 \times 224$ and $32 \times 32$ resolutions for Transformer and CNN architectures. We only changed the batch size to 100 for CIFAR-100 and borrowed other settings the same as DeiT on ImageNet-1K.

Table A1: Ingredients and hyper-parameters used for ImageNet-1K training settings.

| Procedure | PyTorch | DeiT | RSB A2 | RSB A3 |
|---|---|---|---|---|
| Train Res | 224 | 224 | 224 | 160 |
| Test Res | 224 | 224 | 224 | 224 |
| Test crop ratio | 0.875 | 0.875 | 0.95 | 0.95 |
| Epochs | 100/300 | 300 | 300 | 100 |
| Batch size | 256 | 1024 | 2048 | 2048 |
| Optimizer | SGD | AdamW | LAMB | LAMB |
| LR | 0.1 | $1 \times 10^{-3}$ | $5 \times 10^{-3}$ | $8 \times 10^{-3}$ |
| LR decay | cosine | cosine | cosine | cosine |
| Weight decay | $10^{-4}$ | 0.05 | 0.02 | 0.02 |
| Warmup epochs | ✗ | 5 | 5 | 5 |
| Label smoothing $\epsilon$ | ✗ | 0.1 | ✗ | ✗ |
| Dropout | ✗ | ✗ | ✗ | ✗ |
| Stoch. Depth | ✗ | 0.1 | 0.05 | ✗ |
| Repeated Aug | ✗ | ✓ | ✓ | ✗ |
| Gradient Clip. | ✗ | 1.0 | ✗ | ✗ |
| H. flip | ✓ | ✓ | ✓ | ✓ |
| RRC | ✓ | ✓ | ✓ | ✓ |
| Rand Augment | ✗ | 9/0.5 | 7/0.5 | 6/0.5 |
| Auto Augment | ✗ | ✗ | ✗ | ✗ |
| Mixup alpha | ✗ | 0.8 | 0.1 | 0.1 |
| Cutmix alpha | ✗ | 1.0 | 1.0 | 1.0 |
| Erasing prob. | ✗ | 0.25 | ✗ | ✗ |
| ColorJitter | ✗ | ✗ | ✗ | ✗ |
| EMA | ✗ | ✓ | ✗ | ✗ |
| CE loss | ✓ | ✓ | ✗ | ✗ |
| BCE loss | ✗ | ✗ | ✓ | ✓ |
| Mixed precision | ✗ | ✗ | ✓ | ✓ |

Table A2: Top-1 accuracy (%) of image classification based on ResNet variants on ImageNet-1K using PyTorch-style 100-epoch and 300-epoch training procedures.

| Methods | Beta $\alpha$ | PyTorch 100 epochs | | | | | PyTorch 300 epochs | | | |
|---|---|---|---|---|---|---|---|---|---|---|
| | | R-18 | R-34 | R-50 | R-101 | RX-101 | R-18 | R-34 | R-50 | R-101 |
| Vanilla | - | 70.04 | 73.85 | 76.83 | 78.18 | 78.71 | 71.83 | 75.29 | 77.35 | 78.91 |
| MixUp | 0.2 | 69.98 | 73.97 | 77.12 | 78.97 | 79.98 | 71.72 | 75.73 | 78.44 | 80.60 |
| CutMix | 1 | 68.95 | 73.58 | 77.17 | 78.96 | 80.42 | 71.01 | 75.16 | 78.69 | 80.59 |
| ManifoldMix | 0.2 | 69.98 | 73.98 | 77.01 | 79.02 | 79.93 | 71.73 | 75.44 | 78.21 | 80.64 |
| SaliencyMix | 1 | 69.16 | 73.56 | 77.14 | 79.32 | 80.27 | 70.21 | 75.01 | 78.46 | 80.45 |
| FMix | 1 | 69.96 | 74.08 | 77.19 | 79.09 | 80.06 | 70.30 | 75.12 | 78.51 | 80.20 |
| ResizeMix | 1 | 69.50 | 73.88 | 77.42 | 79.27 | 80.55 | 71.32 | 75.64 | 78.91 | 80.52 |
| PuzzleMix | 1 | 70.12 | 74.26 | 77.54 | 79.43 | 80.53 | 71.64 | 75.84 | 78.86 | 80.67 |
| AutoMix | 2 | 70.50 | 74.52 | 77.91 | 79.87 | 80.89 | 72.05 | 76.10 | 79.25 | 80.98 |
| AdAutoMix | 1 | 70.86 | 74.82 | 78.04 | 79.91 | **81.09** | - | - | - | - |
| SAMix | 2 | **70.83** | **74.95** | **78.06** | **80.05** | 80.98 | **72.27** | **76.28** | **79.39** | **81.10** |

## B  Mixup Image Classification Benchmarks

### B.1  Mixup Benchmarks on ImageNet-1k

**PyTorch-style training settings**  The benchmark results are illustrated in Table A2. Notice that we adopt $\alpha = 0.2$ for some cutting-based mixups (CutMix, SaliencyMix, FMix, ResizeMix) based on ResNet-18 since ResNet-18 might be under-fitted on ImageNet-1k.

**DeiT training setting**  Table A3 shows the benchmark results following DeiT training setting. Experiment details refer to Sec. A.2. Notice that the performances of transformer-based architectures are more difficult to reproduce than ResNet variants, and the mean of the best performance in 3 trials is reported as their original paper.

**RSB A2/A3 training settings**  The RSB A2/A3 benchmark results based on ResNet-50, EfficientNet-B0, and MobileNet.V2 are illustrated in Table A4. Training 300/100 epochs with the BCE loss on ImageNet-1k, RSB A3 is a fast training setting, while RSB A2 can exploit the full representation ability of ConvNets. Notice that the RSB settings employ Mixup with $\alpha = 0.1$ and CutMix with $\alpha = 1.0$. We report the mean of top-1 accuracy in the last 5/10 training epochs for 100/300 epochs.

Table A3: Top-1 accuracy (%) on ImageNet-1K based on popular Transformer-based architectures using DeiT-S training settings. Notice that † denotes reproducing results with the official implementation, while other results are implemented with OpenMixup. TransMix, TokenMix, and SMMix are specially designed for Transformers.

| Methods | $\alpha$ | DeiT-T | DeiT-S | DeiT-B | PVT-T | PVT-S | Swin-T | ConvNeXt-T | MogaNet-T |
|---|---|---|---|---|---|---|---|---|---|
| Vanilla | - | 73.91 | 75.66 | 77.09 | 74.67 | 77.76 | 80.21 | 79.22 | 79.25 |
| DeiT | 0.8, 1 | 74.50 | 79.80 | 81.83 | 75.10 | 78.95 | 81.20 | 82.10 | 79.02 |
| MixUp | 0.2 | 74.69 | 77.72 | 78.98 | 75.24 | 78.69 | 81.01 | 80.88 | 79.29 |
| CutMix | 0.2 | 74.23 | 80.13 | 81.61 | 75.53 | 79.64 | 81.23 | 81.57 | 78.37 |
| ManifoldMix | 0.2 | - | - | - | - | - | - | 80.57 | 79.07 |
| AttentiveMix+ | 2 | 74.07 | 80.32 | 82.42 | 74.98 | 79.84 | 81.29 | 81.14 | 77.53 |
| SaliencyMix | 0.2 | 74.17 | 79.88 | 80.72 | 75.71 | 79.69 | 81.37 | 81.33 | 78.74 |
| FMix | 0.2 | 74.41 | 77.37 | | 75.28 | 78.72 | 79.60 | 81.04 | 79.05 |
| ResizeMix | 1 | 74.79 | 78.61 | 80.89 | 76.05 | 79.55 | 81.36 | 81.64 | 78.77 |
| PuzzleMix | 1 | 73.85 | 80.45 | 81.63 | 75.48 | 79.70 | 81.47 | 81.48 | 78.12 |
| AutoMix | 2 | 75.52 | 80.78 | 82.18 | 76.38 | 80.64 | 81.80 | 82.28 | 79.43 |
| SAMix | 2 | **75.83** | **80.94** | 82.85 | **76.60** | 80.78 | **81.87** | **82.35** | **79.62** |
| TransMix | 0.8, 1 | 74.56 | 80.68 | 82.51 | 75.50 | 80.50 | 81.80 | - | - |
| TokenMix† | 0.8, 1 | 75.31 | 80.80 | **82.90** | 75.60 | - | 81.60 | - | - |
| SMMix | 0.8, 1 | 75.56 | 81.10 | 82.90 | 75.60 | **81.03** | 81.80 | - | - |

Table A4: Top-1 accuracy (%) on ImageNet-1K based on classical ConvNets using RSB A2/A3 training settings, including ResNet, EfficientNet, and MobileNet.V2.

| Backbones Settings | $Beta$ $\alpha$ | R-50 A3 | R-50 A2 | Eff-B0 A3 | Eff-B0 A2 | Mob.V2 $1\times$ A3 | Mob.V2 $1\times$ A2 |
|---|---|---|---|---|---|---|---|
| RSB | 0.1, 1 | 78.08 | 79.80 | 74.02 | 77.26 | 69.86 | 72.87 |
| MixUp | 0.2 | 77.66 | 79.39 | 73.87 | 77.19 | 70.17 | 72.78 |
| CutMix | 0.2 | 77.62 | 79.38 | 73.46 | 77.24 | 69.62 | 72.23 |
| ManifoldMix | 0.2 | 77.78 | 79.47 | 73.83 | 77.22 | 70.05 | 72.34 |
| AttentiveMix+ | 2 | 77.46 | 79.34 | 72.16 | 75.95 | 67.32 | 70.30 |
| SaliencyMix | 0.2 | 77.93 | 79.42 | 73.42 | 77.67 | 69.69 | 72.07 |
| FMix | 0.2 | 77.76 | 79.05 | 73.71 | 77.33 | 70.10 | 72.79 |
| ResizeMix | 1 | 77.85 | 79.94 | 73.67 | 77.27 | 69.94 | 72.50 |
| PuzzleMix | 1 | 78.02 | 79.78 | 74.10 | 77.35 | 70.04 | 72.85 |
| AutoMix | 2 | 78.44 | 80.28 | 74.61 | 77.58 | 71.16 | 73.19 |
| SAMix | 2 | **78.64** | **80.40** | **75.28** | **77.69** | **71.24** | **73.42** |

## B.2 Small-scale Classification Benchmarks

To facilitate fast research on mixup augmentations, we benchmark mixup image classification on CIFAR-10/100 and Tiny-ImageNet with two settings.

**CIFAR-10** As elucidated in Sec. A.2, CIFAR-10 benchmarks based on CIFAR version ResNet variants follow CutMix settings, training 200/400/800/1200 epochs from stretch. As shown in Table A5, we report the median of top-1 accuracy in the last 10 training epochs.

**CIFAR-100** As for the classical setting (a), CIFAR-100 benchmarks train 200/400/800/1200 epochs from the stretch in Table A6, which is similar to CIFAR-10. Notice that we set weight decay to 0.0005 for cutting-based methods (CutMix, AttentiveMix+, SaliencyMix, FMix, ResizeMix) for better performances when using ResNeXt-50 (32x4d) as the backbone. As shown in Table A7 using the modern setting (b), we train three modern architectures for 200/600 epochs from the stretch. We resize the raw images to $224 \times 224$ resolutions for DeiT-S and Swin-T, while modifying the stem network as the CIFAR version of ResNet for ConvNeXt-T with $32 \times 32$ resolutions. As shown in Table A8, we further provided more metrics to evaluate the robustness (top-1 accuracy on the corrupted version of CIFAR-100 [64] and applying FGSM attack [61]) and the prediction calibration.

**Tiny-ImageNet** We largely follow the training setting of PuzzleMix [11] on Tiny-ImageNet, which adopts the basic augmentations of `RandomFlip` and `RandomResizedCrop` and optimize the models with a basic learning rate of 0.2 for 400 epochs with Cosine Scheduler. As shown in Table A9, all compared methods adopt ResNet-18 and ResNeXt-50 (32x4d) architectures training 400 epochs from the stretch on Tiny-ImageNet.

Table A5: Top-1 accuracy (%) on CIFAR-10 training 200, 400, 800, 1200 epochs based on ResNet (R) and ResNeXt-32x4d (RX).

| Backbones | Beta | R-18 | R-18 | R-18 | R-18 | Beta | RX-50 | RX-50 | RX-50 | RX-50 |
|---|---|---|---|---|---|---|---|---|---|---|
| Epochs | $\alpha$ | 200 ep | 400 ep | 800 ep | 1200ep | $\alpha$ | 200 ep | 400 ep | 800 ep | 1200ep |
| Vanilla | - | 94.87 | 95.10 | 95.50 | 95.59 | - | 95.92 | 95.81 | 96.23 | 96.26 |
| MixUp | 1 | 95.70 | 96.55 | 96.62 | 96.84 | 1 | 96.88 | 97.19 | 97.30 | 97.33 |
| CutMix | 0.2 | 96.11 | 96.13 | 96.68 | 96.56 | 0.2 | 96.78 | 96.54 | 96.60 | 96.35 |
| ManifoldMix | 2 | 96.04 | 96.57 | 96.71 | 97.02 | 2 | 96.97 | 97.39 | 97.33 | 97.36 |
| SmoothMix | 0.5 | 95.29 | 95.88 | 96.17 | 96.17 | 0.2 | 95.87 | 96.37 | 96.49 | 96.77 |
| AttentiveMix+ | 2 | 96.21 | 96.45 | 96.63 | 96.49 | 2 | 96.84 | 96.91 | 96.87 | 96.62 |
| SaliencyMix | 0.2 | 96.05 | 96.42 | 96.20 | 96.18 | 0.2 | 96.65 | 96.89 | 96.70 | 96.60 |
| FMix | 0.2 | 96.17 | 96.53 | 96.18 | 96.01 | 0.2 | 96.72 | 96.76 | 96.76 | 96.10 |
| GridMix | 0.2 | 95.89 | 96.33 | 96.56 | 96.58 | 0.2 | 97.18 | 97.30 | 96.40 | 95.79 |
| ResizeMix | 1 | 96.16 | 96.91 | 96.76 | 97.04 | 1 | 97.02 | 97.38 | 97.21 | 97.36 |
| PuzzleMix | 1 | 96.42 | 96.87 | 97.10 | 97.13 | 1 | 97.05 | 97.24 | 97.37 | 97.34 |
| AutoMix | 2 | 96.59 | 97.08 | 97.34 | 97.30 | 2 | 97.19 | 97.42 | 97.65 | 97.51 |
| SAMix | 2 | **96.67** | **97.16** | **97.50** | **97.41** | 2 | **97.23** | **97.51** | **97.93** | **97.74** |

Table A6: Top-1 accuracy (%) on CIFAR-100 training 200, 400, 800, 1200 epochs based on ResNet (R), Wide-ResNet (WRN), ResNeXt-32x4d (RX). Notice that † denotes reproducing results with the official implementation, while other results are implemented with OpenMixup.

| Backbones | Beta | R-18 | R-18 | R-18 | R-18 | RX-50 | RX-50 | RX-50 | RX-50 | WRN-28-8 |
|---|---|---|---|---|---|---|---|---|---|---|
| Epochs | $\alpha$ | 200 ep | 400 ep | 800 ep | 1200ep | 200 ep | 400 ep | 800 ep | 1200ep | 400ep |
| Vanilla | - | 76.42 | 77.73 | 78.04 | 78.55 | 79.37 | 80.24 | 81.09 | 81.32 | 81.63 |
| MixUp | 1 | 78.52 | 79.34 | 79.12 | 79.24 | 81.18 | 82.54 | 82.10 | 81.77 | 82.82 |
| CutMix | 0.2 | 79.45 | 79.58 | 78.17 | 78.29 | 81.52 | 78.52 | 78.32 | 77.17 | 84.45 |
| ManifoldMix | 2 | 79.18 | 80.18 | 80.35 | 80.21 | 81.59 | 82.56 | 82.88 | 83.28 | 83.24 |
| SmoothMix | 0.2 | 77.90 | 78.77 | 78.69 | 78.38 | 80.68 | 79.56 | 78.95 | 77.88 | 82.09 |
| SaliencyMix | 0.2 | 79.75 | 79.64 | 79.12 | 77.66 | 80.72 | 78.63 | 78.77 | 77.51 | 84.35 |
| AttentiveMix+ | 2 | 79.62 | 80.14 | 78.91 | 78.41 | 81.69 | 81.53 | 80.54 | 79.60 | 84.34 |
| FMix | 0.2 | 78.91 | 79.91 | 79.69 | 79.50 | 79.87 | 78.99 | 79.02 | 78.24 | 84.21 |
| GridMix | 0.2 | 78.23 | 78.60 | 78.72 | 77.58 | 81.11 | 79.80 | 78.90 | 76.11 | 84.24 |
| ResizeMix | 1 | 79.56 | 79.19 | 80.01 | 79.23 | 79.56 | 79.78 | 80.35 | 79.73 | 84.87 |
| PuzzleMix | 1 | 79.96 | 80.82 | 81.13 | 81.10 | 81.69 | 82.84 | 82.85 | 82.93 | 85.02 |
| Co-Mixup† | 2 | 80.01 | 80.87 | 81.17 | 81.18 | 81.73 | 82.88 | 82.91 | 82.97 | 85.05 |
| AutoMix | 2 | 80.12 | 81.78 | 82.04 | 81.95 | 82.84 | 83.32 | 83.64 | 83.80 | 85.18 |
| SAMix | 2 | 81.21 | **81.97** | 82.30 | **82.41** | 83.81 | **84.27** | **84.42** | **84.31** | **85.50** |
| AdAutoMix | 1 | **81.55** | **81.97** | **82.32** | - | **84.40** | 84.05 | 84.42 | - | 85.32 |

Table A7: Top-1 accuracy (%), GPU memory (G), and total training time (h) of 600 epochs on CIFAR-100 training 200 and 600 epochs based on DeiT-S, Swin-T, and ConvNeXt-T with the DeiT training setting. Notice that all methods are trained on a single A100 GPU to collect training times and GPU memory.

| Methods | $\alpha$ | DeiT-Small | | | | Swin-Tiny | | | | ConvNeXt-Tiny | | | |
|---|---|---|---|---|---|---|---|---|---|---|---|---|---|
| | | 200 ep | 600 ep | Mem. | Time | 200 ep | 600 ep | Mem. | Time | 200 ep | 600 ep | Mem. | Time |
| Vanilla | - | 65.81 | 68.50 | 8.1 | 27 | 78.41 | 81.29 | 11.4 | 36 | 78.70 | 80.65 | 4.2 | 10 |
| Mixup | 0.8 | 69.98 | 76.35 | 8.2 | 27 | 76.78 | 83.67 | 11.4 | 36 | 81.13 | 83.08 | 4.2 | 10 |
| CutMix | 2 | 74.12 | 79.54 | 8.2 | 27 | 80.64 | 83.38 | 11.4 | 36 | 82.46 | 83.20 | 4.2 | 10 |
| DeiT | 0.8, 1 | 75.92 | 79.38 | 8.2 | 27 | 81.25 | 84.41 | 11.4 | 36 | 83.09 | 84.12 | 4.2 | 10 |
| ManifoldMix | 2 | - | - | 8.2 | 27 | - | - | 11.4 | 36 | 82.06 | 83.94 | 4.2 | 10 |
| SmoothMix | 0.2 | 67.54 | 80.25 | 8.2 | 27 | 66.69 | 81.18 | 11.4 | 36 | 78.87 | 81.31 | 4.2 | 10 |
| SaliencyMix | 0.2 | 69.78 | 76.60 | 8.2 | 27 | 80.40 | 82.58 | 11.4 | 36 | 82.82 | 83.03 | 4.2 | 10 |
| AttentiveMix+ | 2 | 75.98 | 80.33 | 8.3 | 35 | 81.13 | 83.69 | 11.5 | 43 | 82.59 | 83.04 | 4.3 | 14 |
| FMix | 1 | 70.41 | 74.31 | 8.2 | 27 | 80.72 | 82.82 | 11.4 | 36 | 81.79 | 82.29 | 4.2 | 10 |
| GridMix | 1 | 68.86 | 74.96 | 8.2 | 27 | 78.54 | 80.79 | 11.4 | 36 | 79.53 | 79.66 | 4.2 | 10 |
| ResizeMix | 1 | 68.45 | 71.95 | 8.2 | 27 | 80.16 | 82.36 | 11.4 | 36 | 82.53 | 82.91 | 4.2 | 10 |
| PuzzleMix | 2 | 73.60 | 81.01 | 8.3 | 35 | 80.33 | 84.74 | 11.5 | 45 | 82.29 | 84.17 | 4.3 | 53 |
| AlignMix | 1 | - | - | - | - | 78.91 | 83.34 | 12.6 | 39 | 80.88 | 83.03 | 4.2 | 13 |
| AutoMix | 2 | 76.24 | 80.91 | 18.2 | 59 | 82.67 | 84.05 | 29.2 | 75 | 83.30 | 84.79 | 10.2 | 56 |
| SAMix | 2 | **77.94** | **82.49** | 21.3 | 58 | **82.70** | **84.74** | 29.3 | 75 | **83.56** | **84.98** | 10.3 | 57 |
| TransMix | 0.8, 1 | 76.17 | 79.33 | 8.4 | 28 | 81.33 | 84.45 | 11.5 | 37 | - | - | - | - |
| SMMix | 0.8, 1 | 74.49 | 80.05 | 8.4 | 28 | 81.55 | - | 11.5 | 37 | - | - | - | - |
| Decoupled (DeiT) | 0.8, 1 | 76.75 | 79.78 | 8.2 | 27 | 81.10 | 84.59 | 11.4 | 36 | 83.44 | 84.49 | 4.2 | 10 |

Table A8: More evaluation metric (robustness and calibration) on CIFAR-100 with 200-epoch training, reporting top-1 accuracy (%)↑ (clean data, corruption data, and FGSM attacks) and calibration ECE (%)↓.

| Methods | $\alpha$ | DeiT-Small | | | | Swin-Tiny | | | |
|---|---|---|---|---|---|---|---|---|---|
| | | Clean | Corruption | FGSM | ECE↓ | Clearn | Corruption | FGSM | ECE↓ |
| Vanilla | - | 65.81 | 49.31 | 20.58 | 9.48 | 78.41 | 58.20 | 12.87 | 11.67 |
| Mixup | 0.8 | 69.98 | 55.85 | 17.65 | 7.38 | 76.78 | 59.11 | 15.03 | 13.89 |
| CutMix | 2 | 74.12 | 55.08 | 12.53 | 6.18 | 80.64 | 57.73 | 18.38 | 10.95 |
| DeiT | 0.8, 1 | 75.92 | 57.36 | 18.55 | 5.38 | 81.25 | 62.21 | 15.66 | 15.68 |
| SmoothMix | 0.2 | 67.54 | 52.42 | 15.07 | 30.59 | 66.69 | 49.69 | 9.79 | 27.10 |
| SaliencyMix | 0.2 | 69.78 | 51.14 | 17.31 | 5.45 | 80.40 | 58.43 | 15.29 | 10.49 |
| AttentiveMix+ | 2 | 75.98 | 57.57 | 13.90 | 9.89 | 81.13 | 58.07 | 15.43 | 9.60 |
| FMix | 1 | 70.41 | 51.94 | 12.20 | 4.14 | 80.72 | 58.44 | 13.97 | 9.19 |
| GridMix | 1 | 68.86 | 51.11 | 8.43 | 4.09 | 78.54 | 57.78 | 11.07 | 9.37 |
| ResizeMix | 1 | 68.45 | 50.87 | 20.03 | 7.64 | 80.16 | 57.37 | 13.64 | 7.68 |
| PuzzleMix | 2 | 73.60 | 57.67 | 17.44 | 9.45 | 80.33 | 60.67 | 12.96 | 16.23 |
| AlignMix | 1 | - | - | - | - | 78.91 | 61.61 | 17.20 | **1.92** |
| AutoMix | 2 | 76.24 | 60.08 | 27.35 | 4.69 | 82.67 | **64.10** | 23.62 | 9.19 |
| SAMix | 2 | **77.94** | **61.91** | **30.35** | **4.01** | **82.70** | 62.19 | **23.66** | 7.85 |
| TransMix | 0.8, 1 | 76.17 | 59.89 | 22.48 | 8.28 | 81.33 | 62.53 | 18.90 | 16.47 |
| SMMix | 0.8, 1 | 74.49 | 59.96 | 22.85 | 8.34 | 81.55 | 62.86 | 19.14 | 16.81 |
| Decoupled (DeiT) | 0.8, 1 | 76.75 | 59.89 | 22.48 | 8.28 | 81.10 | 62.25 | 16.54 | 16.16 |

Table A9: Top-1 accuracy (%) on Tiny based on ResNet (R) and ResNeXt-32x4d (RX). Notice that † denotes reproducing results with the official implementation, while other results are implemented with OpenMixup.

| Backbones | $\alpha$ | R-18 | RX-50 |
|---|---|---|---|
| Vanilla | - | 61.68 | 65.04 |
| MixUp | 1 | 63.86 | 66.36 |
| CutMix | 1 | 65.53 | 66.47 |
| ManifoldMix | 0.2 | 64.15 | 67.30 |
| SmoothMix | 0.2 | 66.65 | 69.65 |
| AttentiveMix+ | 2 | 64.85 | 67.42 |
| SaliencyMix | 1 | 64.60 | 66.55 |
| FMix | 1 | 63.47 | 65.08 |
| GridMix | 0.2 | 65.14 | 66.53 |
| ResizeMix | 1 | 63.74 | 65.87 |
| PuzzleMix | 1 | 65.81 | 67.83 |
| Co-Mixup† | 2 | 65.92 | 68.02 |
| AutoMix | 2 | 67.33 | 70.72 |
| SAMix | 2 | 68.89 | 72.18 |
| AdAutoMix | 1 | **69.19** | **72.89** |

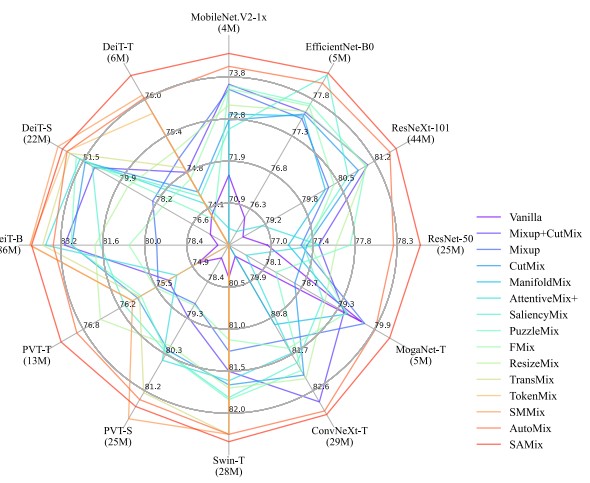

Figure A2: Radar plots of the top-1 accuracy of all evaluated mixup augmentation methods based on a variety of popular vision backbones on ImageNet-1K.

### B.3 Downstream Classification Benchmarks

We further provide benchmarks on three downstream classification scenarios in 224×224 resolutions with ResNet architectures, as shown in Table .

**Benchmarks on Fine-grained Scenarios.** As for fine-grained scenarios, each class usually has limited samples and is only distinguishable in some particular regions. We conduct (a) transfer learning on CUB-200 and FGVC-Aircraft, and (b) fine-grained classification with training from scratch on iNat2017 and iNat2018. For (a), we use transfer learning settings on fine-grained datasets, using PyTorch official pre-trained models as initialization and training 200 epochs by SGD optimizer with the initial learning rate of 0.001, the weight decay of 0.0005, the batch size of 16, the same data augmentation as ImageNet-1K settings. For (b) and (c), we follow Pytorch-style ImageNet-1K settings mentioned above, training 100 epochs from the stretch.

**Benchmarks on Scenis Scenarios.** As for scenic classification tasks, we study whether mixup augmentations help models distinguish the backgrounds, which are less important than the foreground objects in commonly used datasets. We employ the PyTorch-style ImageNet-1K setting on Places205, training models for 100 epochs with SGD optimizer, a basic learning rate of 0.1 with 256 batch size.

Table A10: Top-1 accuracy (%) of mixup image classification with ResNet (R) and ResNeXt (RX) variants on fine-grained datasets (CUB-200, FGVC-Aircraft, iNat2017/2018) and Places205.

| Method | Beta $\alpha$ | CUB-200 R-18 | RX-50 | FGVC-Aircraft R-18 | RX-50 | Beta $\alpha$ | iNat2017 R-50 | RX-101 | iNat2018 R-50 | RX-101 | Beta $\alpha$ | Places205 R-18 | R-50 |
|---|---|---|---|---|---|---|---|---|---|---|---|---|---|
| Vanilla | - | 77.68 | 83.01 | 80.23 | 85.10 | - | 60.23 | 63.70 | 62.53 | 66.94 | - | 59.63 | 63.10 |
| MixUp | 0.2 | 78.39 | 84.58 | 79.52 | 85.18 | 0.2 | 61.22 | 66.27 | 62.69 | 67.56 | 0.2 | 59.33 | 63.01 |
| CutMix | 1 | 78.40 | 85.68 | 78.84 | 84.55 | 1 | 62.34 | 67.59 | 63.91 | 69.75 | 0.2 | 59.21 | 63.75 |
| ManifoldMix | 0.5 | 79.76 | 86.38 | 80.68 | 86.60 | 0.2 | 61.47 | 66.08 | 63.46 | 69.30 | 0.2 | 59.46 | 63.23 |
| SaliencyMix | 0.2 | 77.95 | 83.29 | 80.02 | 84.31 | 1 | 62.51 | 67.20 | 64.27 | 70.01 | 0.2 | 59.50 | 63.33 |
| FMix | 0.2 | 77.28 | 84.06 | 79.36 | 86.23 | 1 | 61.90 | 66.64 | 63.71 | 69.46 | 0.2 | 59.51 | 63.63 |
| ResizeMix | 1 | 78.50 | 84.77 | 78.10 | 84.0 | 1 | 62.29 | 66.82 | 64.12 | 69.30 | 1 | 59.66 | 63.88 |
| PuzzleMix | 1 | 78.63 | 84.51 | 80.76 | 86.23 | 1 | 62.66 | 67.72 | 64.36 | 70.12 | 1 | 59.62 | 63.91 |
| AutoMix | 2 | 79.87 | 86.56 | 81.37 | 86.72 | 2 | 63.08 | 68.03 | 64.73 | 70.49 | 2 | 59.74 | 64.06 |
| SAMix | 2 | **81.11** | **86.83** | **82.15** | **86.80** | 2 | **63.32** | **68.26** | **64.84** | **70.54** | 2 | **59.86** | 64.27 |

Table A11: Trasfer learning of object detection with ImageNet-1k pre-trained ResNet-50 backbone on COCO dataset.

Table A12: Trasfer learning of object detection with Mask R-CNN and semantic segmentation with Semantic FPN with pre-trained PVT-S on COCO and ADE20K, respectively.

| Method | IN-1K Acc | COCO mAP | $AP_{50}^{bb}$ | $AP_{75}^{bb}$ |
|---|---|---|---|---|
| Vanilla | 76.8 | 38.1 | 59.1 | 41.8 |
| Mixup | 77.1 | 37.9 | 59.0 | 41.7 |
| CutMix | 77.2 | 38.2 | 59.3 | 42.0 |
| ResizeMix | 77.4 | 38.4 | 59.4 | 42.1 |
| PuzzleMix | 77.5 | 38.3 | 59.3 | 42.1 |
| AutoMix | 77.9 | 38.6 | 59.5 | **42.2** |
| SAMix | **78.1** | **38.7** | **59.6** | **42.2** |

| Method | IN-1K Acc | COCO mAP | $AP_{50}^{bb}$ | $AP_{75}^{bb}$ | ADE20K mIoU |
|---|---|---|---|---|---|
| MixUp+CutMix | 79.8 | 40.4 | 62.9 | 43.8 | 41.9 |
| AutoMix | 80.7 | 40.9 | 63.9 | 44.1 | 42.5 |
| TransMix | 80.5 | 40.9 | 63.8 | 44.0 | 42.6 |
| TokenMix | 80.6 | **41.0** | **64.0** | 44.3 | **42.7** |
| TokenMixup | 80.5 | 40.7 | 63.6 | 43.9 | 42.5 |
| SMMix | **81.0** | **41.0** | 63.9 | **44.4** | **43.0** |

## B.4 Transfer Learning

**Object Detection.** We conduct transfer learning experiments with pre-trained ResNet-50 [52] and PVT-S [65] using mixup augmentations to object detection on COCO-2017 [46] dataset, which evaluate the generalization abilities of different mixup approaches. We first fine-tune Faster RCNN [44] with ResNet-50-C4 using Detectron2 [66] in Table A11, which is trained by SGD optimizer and multi-step scheduler for 24 epochs (2×). The *dynamic* mixup methods (*e.g.,* AutoMix) usually achieve both competitive performances in classification and object detection tasks. Then, we fine-tune Mask R-CNN [45] by AdamW optimizer for 24 epochs using MMDetection [67] in Table A12. We have integrated Detectron2 and MMDetection into OpenMixup, and the users can perform the transferring experiments with pre-trained models and config files. Compared to *dynamic* sample mixing methods, recently-proposed label mixing policies (*e.g.,* TokenMix and SMMix) yield better performances with less extra training overheads.

**Semantic Segmentation.** We also perform transfer learning to semantic segmentation on ADE20K [47] with Semantic FPN [62] to evaluate the generalization abilities to fine-grained prediction tasks. Following PVT [65], we fine-tuned Semantic FPN for 80K interactions by AdamW [68] optimizer with the learning rate of $2 \times 10^{-4}$ and a batch size of 16 on $512^2$ resolutions using MMSegmentation [69]. Table A12 shows the results of transfer experiments based on PVT-S.

## B.5 Rules for Counting the Mixup Rankings

We have summarized and analyzed a great number of mixup benchmarking results to compare and rank all the included mixup methods in terms of *performance*, *applicability*, and the *overall* capacity. Specifically, regarding the *performance*, we averaged the accuracy rankings of all mixup algorithms for each downstream task and averaged their robustness and calibration results rankings separately. Finally, these ranking results are averaged again to produce a comprehensive range of performance ranking results. As for the *applicability*, we adopt a similar ranking computation scheme considering the *time usage* and the *generalizability* of the methods. With the *overall* capacity ranking, we combined the performance and applicability rankings with a 1:1 weighting to obtain the final take-home rankings. For equivalent results, we take a tied ranking approach. For instance, if three methods are tied for first place, then the method that results in fourth place is recorded as second place by default. Finally, we provide the comprehensive rankings as shown in Table 1 and Table 5.