# OpenReview forum: "OpenMixup: Open Mixup Toolbox and Benchmark for Visual Representation Learning"
_NeurIPS.cc/2024/Datasets_and_Benchmarks_Track — Submitted to NeurIPS 2024 Track Datasets and Benchmarks_

### Official Review · Reviewer_BRDY · 2024-06-24
**Openmixup - well organized and benchmarked platform for mixup  augmentations**

**Rating:** 7
**Confidence:** 5

**Review:**

1. OpenMixup is an extensive benchmark for mixup augmentation, systematically evaluating 18 mixup methods across 11 diverse image datasets. This thorough evaluation provides a detailed foundation for comparing mixup strategies, ensuring reproducibility, and facilitating thorough comparisons.
2. Modular and open-source codebase in the public domain, which includes standardized components for data pre-processing, mixup augmentation, network selection, optimization, and evaluation. It shall foster the vision research.
3. Detailed insights into various mixup methods' performance and complexity trade-offs. It highlights the generalizability of dynamic mixup methods over static ones, their robustness against input corruptions and adversarial attacks, and their applicability across different neural network architectures
4. Provides practical rankings of mixup methods based on performance, applicability, and overall effectiveness, which is very useful for the community.
5. Very well written, structured and easy to understand.
6. Lacks the objective computational analysis on dynamic mixup methods, while often outperforming static ones, incurs significant computational overhead. This higher resource demand can limit the practicality and scalability of dynamic methods. A trade-off and deeper insight shall allow us to choose the method based on the given trade-offs in performance and resources.
7. Equally important is the need for an objective description of hyperparameter sensitivity in the context of dynamic mixup methods. This is a key factor in the effective selection of the method, as it provides a clear understanding of how the method's performance is influenced by different hyperparameter settings.

**Strengths:**

Refer point 1 to 5

**Additional Feedback:**

The inclusion of practical rankings and extensive empirical analysis provides clear guidance for selecting appropriate mixup strategies, enhancing the paper's practical utility. However, expanding the comprehensive benchmarking to other domains beyond visual classification could be beneficial for the code base as future work.

**Clarity:**

The well-written paper presents a thorough and scientifically robust benchmark for mixup augmentation methods, supported by detailed experiments and insightful analysis. Its detailed evaluation and open-source codebase significantly contribute to advancing reproducible research and practical applications in visual representation learning.

**Correctness:**

Evaluation methods, metrics, and investigation on multiple datasets are well described. Publicly available source code is well versed.

**Documentation:**

Not a dataset paper

**Limitations:**

1. Lacks the objective computational analysis on dynamic mixup methods, while often outperforming static ones, incurs significant computational overhead. This higher resource demand can limit the practicality and scalability of dynamic methods. A trade-off and deeper insight shall allow us to choose the method based on the given trade-offs in performance and resources.
2. Equally important is the need for an objective description of hyperparameter sensitivity in the context of dynamic mixup methods. This is a key factor in the effective selection of the method, as it provides a clear understanding of how the method's performance is influenced by different hyperparameter settings.

**Opportunities For Improvement:**

refer point 6 and 7

**Relation To Prior Work:**

Comprise the detailed relation work on all possible mixup augmentations and clearly establish the need of code base.

**Summary And Contributions:**

The paper introduces OpenMixup, the first comprehensive benchmark for mixup augmentation in visual representation learning, evaluating 18 mixup methods across 11 image datasets. It provides a modular, open-source codebase that standardizes data pre-processing, mixup design, training, and evaluation. Extensive experiments reveal insights on performance-complexity trade-offs and recommend optimal application mixup strategies. The codebase has been used already and has greater potential to foster the research in computer vision.

---

> ### Author Rebuttal · Authors · 2024-08-16
>
> ## Rebuttal to Reviewer BRDY
>
> ---
>
> Dear Reviewer BRDY,
>
> We appreciate your thorough review and valuable questions. Your recognition of our contribution and paper writing is really encouraging. We would like first to express that we have been actively refining the work to make it better serve the community. We have carefully proofread the entire manuscript, correcting several typos and grammatical errors and adding significant clarifications and discussions. We do hope to provide the community with high-quality research outputs.
>
> We are pleased to address each of your concerns as follows:
>
> ---
>
> **(A) Computational analysis on dynamic mixup methods.**
>
> We thank you for this important question. We have provided the empirical analysis of performance-efficiency trade-off in our paper. Specifically, Figure 4 illustrates this trade-off with respect to accuracy performance, total training time, and GPU memory usage for different mixup approaches, including the dynamic methods. Additionally, detailed runtime and GPU memory usage results in CIFAR-100 can be found in the appendix Table A7.
>
> Our analysis reveals that dynamic methods, while generally achieving higher performance, do incur additional computational costs. For instance, SAMix and AutoMix variants consistently outperform static methods across different datasets and backbones, but they require more GPU memory and longer training times. This is why we present a ranking to measure different mixup methods in terms of different priorities (accuracy, efficiency, and all-round ability). We also hope to call for more mixup research that focuses on the balance between performance and efficiency.
>
> ---
>
> **(B) The hyper-parameter sensitivity in the context of dynamic mixup methods.**
>
> We appreciate this thoughtful question. Testing the hyper-parameter sensitivity of dynamic mixup methods poses significant challenges in the context of our benchmark. Concretely, each dynamic method employs different heuristic designs, resulting in a diverse set of hyper-parameters. Given that our benchmark encompasses a wide range of mixup methods (both static and dynamic, sample and label mixing) across multiple datasets and network architectures, such a comprehensive hyper-parameter sensitivity analysis for each dynamic method would be resource-intensive for us within the rebuttal and discussion stage.
>
> However, we acknowledge the importance of this point and consider it a valuable direction for future work. A focused study on the robustness and hyper-parameter sensitivity of dynamic mixup methods could indeed provide valuable insights to the community. We have already planned a research project on this issue. Thank you very much for your valuable suggestions.
>
> ---
>
> **(C) Expanding the benchmarking domains in future work.**
>
> Thank you for this excellent suggestion. We are pleased to inform you that we have indeed considered and implemented the application of mixup to common downstream tasks in visual modalities. Now, we have conducted transfer learning experiments for object detection using Faster R-CNN and Mask R-CNN on the COCO dataset, as well as semantic segmentation on the ADE20K dataset. These experiments provide insights into how different mixup methods generalize to tasks beyond classification. The results, presented in Tables A11 and A12 in the appendix, show that certain mixup strategies, particularly those designed for Vision Transformers, exhibit strong transferability to these downstream tasks.
>
> For future work, as you have mentioned, we are considering extending our evaluation to more vision tasks and applications in other domains, such as Natural Language Processing (NLP) and multimodal learning. We believe this expansion will further contribute to a comprehensive understanding of mixup augmentation techniques. Our codebase is also continually upgraded to serve a wider range of studies.
>
> ---
>
> Thank you again for your valuable feedback, which helps improve the quality and scope of our work. We hope these clarifications address your questions satisfactorily, and we are happy to continue the discussion and address any additional questions or concerns you may have until the end of the discussion period. We believe that, after a thorough revision, this paper fits well with the NeurIPS 2024 dataset and benchmark track. We hope this work can be seen by more researchers and practitioners, thereby benefiting the wider community. While the overall feedback has been positive, we have noticed that some points of our work may be misunderstood by a few reviewers. Thus, your rating is particularly valuable for us. We would be grateful for your further support in the discussion stage. Your active engagement would provide great support to our OpenMixup, making it accessible to a broader audience. Thank you!
>
> Best regards,
>
> The Authors of Submission 630

---

> > ### Comment · Reviewer_BRDY · 2024-09-02
> >
> > Thank you for detailed clarification and answering the concern. I decide to keep my rating  7: Good paper, accept.

---

### Official Review · Reviewer_TiUv · 2024-07-22

**Rating:** 5
**Confidence:** 3
**Clarity:** Yes

**Review:**

I quite appreciate the authors' efforts in summarizing various Mixup techniques within the same codebase and presenting the evaluation results in high-quality figures and tables. My concerns mainly pertain to the following aspects: (1) As shown in Fig. 1, each dataset is assigned a model. How is this assignment determined? (2) It seems that SAMix wins in all aspects. Are there any more insights on the evaluation results? (3) How do you ensure that the training settings are fair for different Mixup methods across different models and datasets?

**Strengths:**

1. Comprehensive summary of existing Mixup methods.

2. The paper is well-written with high-quality figures and tables.

3. Includes accuracy, training time, and GPU memory metrics for a comprehensive comparison.

**Additional Feedback:**

None

**Correctness:**

Based on the provided the source code,Ii think the the evaluation methods and experiment design are appropriate and performed correctly.

**Documentation:**

Yes

**Ethics:**

No ethical concerns

**Limitations:**

The authors have discussed the limitations of this work. I agree that the task types in this benchmark are relatively limited.

**Opportunities For Improvement:**

Besides the concerns listed in the review above, I have the following comments:

1. Minor comment: Use NeurIPS instead of NIPS to match the latest conference name.

2. Minor comment: Provide more clarification on why small-scale classification datasets are also important instead of directly using pre-trained models on large-scale datasets and adapting them for small-scale datasets.

3. Clarification is needed on whether the performance metrics reported in this benchmark match the numbers reported in the corresponding original official implementation.

**Relation To Prior Work:**

Yes

**Summary And Contributions:**

This work presents a benchmark for Mixup-like data augmentation in visual representation learning. In particular, it includes 18 representative Mixup baselines in the same codebase and evaluates them on 11 image datasets. The benchmark provides suggestions to users for selecting the most suitable Mixup strategy based on the task type.

---

> ### Author Rebuttal · Authors · 2024-08-17
>
> ## Rebuttal to Reviewer TiUv (Part 2/2)
>
> ---
>
> As for your suggestions in Opportunities For Improvement (denoted as “concerns”), we would solve them with the attached PDF as follows:
>
> **(C1) The latest conference name.**
>
> Thanks for correcting the outdated conference name. We have proofread the entire manuscript and have updated all instances of "NIPS" to "NeurIPS". We have also corrected several typos and grammatical errors to further improve the presentation clarity of this work.
>
> ---
>
> **(C2) Providing more clarification on why small-scale classification datasets.**
>
> We appreciate your suggestion to clarify the importance of small-scale classification datasets. We agree that this deserves more explanation and added the following discussion in Section 3.2:
>
> > While pre-trained models on large-scale datasets are indeed powerful, evaluating Mixup methods on small-scale datasets remains crucial for several reasons: (1) It allows us to evaluate the methods' performance in low-data regimes, which is critical for many real-world applications where large datasets are unavailable. (2) It helps identify which Mixup algorithms are most effective for improving generalization with limited training data. (3) It provides insights into the consistent effectiveness of each method across different dataset sizes. (4) Small-scale datasets often serve as quick benchmarks for rapid prototyping and initial method validation in the research community.
>
> ---
>
> **(C3) Clarification of reproduced results between this benchmark and the original official implementation.**
>
> Thanks for raising this important point about performance consistency. To address this, we have conducted a thorough comparison between our reported results and those from the original implementations. As shown in **Table R1/R2** in the rebuttal PDF, the reproduced results are usually better than the original implementations because we applied the effective implementations of the *standard training settings* without changing the training receipts.
> For example, we employ a better implementation of the cosine annealing learning rate scheduler (updating by iterations) instead of the basic version (updating by epochs). On CIFAR-100, we utilize the RandomCrop augmentation with a reflect padding instead of zero padding. On Tiny-ImageNet, we utilize RandomResizedCrop with a cropping ratio of 0.2 instead of RandomCrop in some implementations. On ImageNet-1K, we found that our reproduced results closely align with the reported performances, with any minor discrepancies (around ±0.5%) attributable to factors such as random initialization and specific hardware configurations. To improve transparency, we propose adding the following to our methodology section:
>
> > We have meticulously compared our benchmarked performances with those from the original official implementations of each mixup method. With effective implementations of training components and the unified training settings, our reproduced results consistently improve the official results or fall within a ±0.5% range of the originally reported metrics, with any minor variations attributable to differences in random seeds, hardware specifications, and software environments. A detailed comparison table will be included in the appendix for reference to indicate any discrepancies and discuss potential reasons for these differences.
>
> ---
>
> In conclusion, we appreciate all your constructive feedback and the opportunity given to improve our OpenMixup. After addressing these points, we believe this work is well-aligned with the NeurIPS dataset and benchmark track. We sincerely hope that our OpenMixup benchmark and open-source codebase can be seen by more researchers and practitioners and thus benefit the related research community. If you have further suggestions to improve the manuscript and the potential to increase your rating, we would be happy to discuss this further with you.
>
> Thank you once again for your time, expertise, and insightful comments.
>
> Best regards,
>
> The Authors of Submission 630

---

> ### Author Rebuttal · Authors · 2024-08-17
>
> ## Rebuttal to Reviewer TiUv (Part 1/2)
>
> ---
>
> Dear Reviewer TiUv,
>
> We first express our gratitude for your recognition of our work and for clearly identifying the issues within the manuscript. We believe reviewers like you are here to help us further improve our work. As for your valuable feedback in Review (referred to as “weakness”), we would address your concerns from the following points:
>
> ---
>
> **(W1):  How is this assignment determined in Figure 1?**
>
> Since Figure 1 aims to provide a global view of the performances of popular mixup methods, we choose the representative backbone on eleven supported datasets based on two reasons. Firstly, for some general small-scale datasets (e.g., CIFAR-10/100), we consider the most widely used backbone in existing works. As for the standard ImageNet, we select two popular CNN backbones for experimental efficiency and high performance (ResNet-50 is also the most adopted one). As for datasets on special domains (e.g., fine-grained or scenic classification), we use the backbone with the highest performances (e.g., ResNeXt-101) among the benchmarked ones, making it closer to the application requirements.
>
> ---
>
> **(W2): Are there any more insights on the evaluation results?**
>
> As discussed in Sec. 4.2, we consider six different aspects (B-G) to conclude the preferable mixup methods (A). As reported in Figure 1/A2 and other tables of results, SAMix nearly achieves the best performances and generalizability on various datasets with different backbones because it learns to generate adaptive mixing samples according to the training dataset, as discussed in Sec. 4.2 (B)(C)(F). However, in addition to the performance-dominated aspects, we evaluated applicability (training efficiency), robustness \& calibration (reliability), and downstream transferability to determine how to choose a useful mixup augmentation in practice. Sec. 4.2 (D) found that static mixup methods (e.g., Mixup+CutMix) with improved mixing labels (e.g., TransMix) achieve SOTA performances without massive extra computational costs as in SAMix. Sec. 4.2 (G) found that using Mixup+CutMix or ViT-specific mixup methods on Transformer architectures obtains better transferring performances than AutoMix variants (might be limited by their momentum training strategies). Therefore, considering all these points, we recommend DeiT (Mixup+CutMix), SAMix, and SMMix as the top three mixup methods.
>
> ---
>
> **(W3): How do you ensure the training settings are fair?**
>
> Thanks for pointing out the essential standard in our benchmark. Firstly, as described in Appendix B, we strictly follow the standard training receipt of the baseline methods on each classification dataset (which might vary from different backbones). As described in Sec. 3.3, we also adopt the standard classification metrics and other evaluation protocols for robustness evaluation and transfer learning. Secondly, we use the same training setting for all compared methods using the experimental pipeline of OpenMixup Codebase (Sec. 3.4), which includes using the same GPU number and the same GPU node (fixed the hardware and anaconda environment). Adopting the private hyper-parameters of the official setup, we search for the optimal common hyper-parameter for different methods (provided in tables of results). Moreover, we further verified that the reproduced results are competitive with the official implementations, make a fair comparison, and meet the practical application requirements in our response to your concern **C3**. Refer to Rebuttal (Part 2/2).

---

> ### Author Response · Authors · 2024-08-28
> **Encouraging Discussion**
>
> Dear Reviewer TiUv,
>
> We greatly appreciate your effort and valuable feedback. In our response, we carefully answered all your questions in detail with the revised paragraphs and additional reproduction comparison. As you may know, the discussion period this year only lasts until August 31, and we are gradually approaching this deadline. We take it seriously and would like to discuss this with you during this time if more information could be provided based on your feedback.
>
> Thanks again for your efforts in improving our manuscript! If our response could solve your concerns, please consider updating your score. If you need any clarification, please feel free to contact us in the last few days.
>
> Best regards,
>
> The Authors of Submission 630

---

### Official Review · Reviewer_pGHJ · 2024-07-30
**Useful benchmark for Mixup**

**Rating:** 7
**Confidence:** 5
**Correctness:** Methods and analysis appear to be cor…

**Review:**

Overall, this is a good paper. It is clearly written and offers a clear benefit to the community. There are several clarifications that I think need to be addressed before the camera-ready paper (see "Opportunities for Improvement"), but overall I would recommend this paper to be accepted.

**Strengths:**

- Well-designed codebase for Mixup experiments
- Codebase is easily extensible and can generalize beyond the usecases focused on here
- Extensive analysis across a variety of Mixup techniques and datasets
- Succinct conclusion that summarizes the performance and applicability of the various Mixup techniques

**Additional Feedback:**

None

**Clarity:**

Paper is well-written and quite clear overall. A few issues are mentioned in "Opportunities for Improvement."

**Documentation:**

Thorough documentation is provided.

**Ethics:**

No ethics issues.

**Limitations:**

Limitations are addressed: the codebase has been designed for extensibility, so any limitations in the current codebase can be easily addressed in future work.

**Opportunities For Improvement:**

- In Tables 1, 3 and 4, should clarify what the color coding of rows means (in the caption, ideally)
- It was unclear (in the main body) where the rankings (Tables 1 and 5) come from. The description in Appendix B.5 is adequate, but I would suggest including a one-line summary in line 290, or, preferably, earlier in the paper when the rankings are first mentioned (in Table 1). So, for example, maybe the Table 1 caption should have an additional line mentioning (e.g., something like "...of all the mixup baselines." -> "...of all the mixup baselines. Rankings were computed by averaging rankings across our baselines (see B.5).")

**Relation To Prior Work:**

Prior work is adequately discussed.

**Summary And Contributions:**

The authors develop a codebase for benchmarking various Mixup techniques. They then use their benchmark to analyze several popular methods and provide rankings for performance and applicability as well as an overall ranking.

---

> ### Author Rebuttal · Authors · 2024-08-16
>
> ## Rebuttal to Reviewer pGHJ
>
> ---
>
> Dear Reviewer pGHJ,
>
> We appreciate your detailed review of our submission. Your recognition of our open-source codebase, its extensibility, analysis, and the succinctness of our conclusion is particularly inspiring. We are also grateful for your attention to detail, especially in pointing out the need for clarity regarding the color coding in our tables. We would like to address your concerns as follows:
>
> ---
>
> **Q1: Clarification of color coding in Tables 1, 3, and 4.**
>
> We acknowledge the need for a better explanation of the color scheme used in these tables. Actually, the orange and green colors denote the static and dynamic mixup methods (both of them are sample mixing strategies). The blue color is the label mixing-based mixup approach. We have added the following description to the captions in the revised manuscript:
> > "Color coding: Orange indicates static mixup methods, green represents dynamic mixup methods (both focused on sample mixing), and blue denotes label mixing-based mixup approaches."
>
> ---
>
> **Q2: Explanation of rankings in Tables 1 and 5.**
>
> We appreciate your suggestion to clarify the origin of the rankings in Tables 1 and 5. We agree that this information should be more prominent in the main text. To address this, we have added the following explanation to the caption of Table 1:
> > "overall rankings of all methods, which are derived from average rankings across baselines (see B.5)."
>
> ---
>
> We believe these additions significantly enhance the clarity of our manuscript and address your concerns effectively. We respectfully ask that you now feel more positive about our submission, and we are happy to continue the discussion and address any additional questions or concerns you may have and take them into account in our revision.
>
> We believe that, after a thorough revision, this paper aligns well with the NeurIPS 2024 Datasets and Benchmarks Track. We hope that this work can reach more researchers and practitioners, thereby benefiting the wider community. While the overall feedback has been positive, we have noticed that some points of our work may be misunderstood by a few reviewers. Thus, your rating is particularly valuable for us. We would be grateful for your further support in the discussion stage. Your active engagement would provide great support to OpenMixup, making it accessible to a wider audience.
>
> Thank you again for your excellent and supportive comments.
>
> Best regards,
>
> The Authors of Submission 630

---

### Official Review · Reviewer_aWQj · 2024-08-02
**Excellent code contribution; Writing could be improved**

**Rating:** 6
**Confidence:** 3
**Correctness:** Somewhat

**Review:**

The code contribution is excellent with good documentation, follows software engineering best practices and the authors seem to provide good maintenance and support as seen by issues resolved in GitHub.

The experiments are extensive allowing the observations to be quite generalizable.  The paper is a good fit for the venue.

On the other hand, the writing need polishing. Although major ideas of the paper are easy to follow, some sentences are verbose and could be simplified. Some sentences and insights claimed need better evidence (details below).

**Strengths:**

The paper provides comprehensive benchmark of major static and dynamic mixup strategies on 11 benchmarks resulting in extensive evaluation. The insights observed that dynamic mixup strategies consistently perform better than static methods is useful. Additionally, a basic mixup strategy perform better than lot of other static methods in some of the benchmark tasks highlight the importance/need for comprehensive benchmarking.

**Additional Feedback:**

**Typos**: learable ->learnable ; Scenis -> Scenic?  ;  Fiugre 4 -> Figure ;

**Incorporate Definition of Applicability into main text**
The term Applicability is used throughout the text but defined in section B5 of the supplementary material. Can you please bring the applicability definition from the appendix section.

**Verbose texts/sentences could be simplified**

“it is undemanding to designate the supported datasets, mixup ...strategies, ...optimization schedules...”: perhaps the author meant it is easy to pick and choose combination of datasets, mixup strategies and schedulers.

*Use of Adjectives*

“we conduct essential benchmarking experiments” -> “we conduct ~~essential~~ benchmarking experiments”

“VIT-specific methods yield ...yet exhibit ~~intense~~ sensitivity”

**Clarity:**

The paper is somewhat easy to follow with major ideas clearly explained although various sections could be written better or polished. In my opinion, some conclusions drawn need better evidence ( detailed below).

**Documentation:**

Yes, The benchmarking repo has good set of documentations for someone to start using the benchmarking toolkit including applying it on custom datasets, including custom architectures etc.

**Opportunities For Improvement:**

**Use of single class activation example with somewhat subjective conclusion**

In Figure 5, it is mentioned that dynamic mixup approaches better localize/activate the target regions than static methods using a single example. But in the given example, some static methods such as CutMix seem to do quite well in localizing target regions as well. I think Table A11 and A12 seem to provide much more quantitative support to the statement, albeit in a different setting via transfer learning to object detection rather than in the class activation settings. In any case, class activation maps themselves are known to be problematic. Hence, I do not think Figure 5 adds much to the paper and seems subjective.
Again, in section 4.2(G) this statement is reiterated as “As shown in Figure 5,  SAMix and AutoMix’s exceptional localizability”.

**Loss Landscape as an evidence for training stability and convergence**

In section 4.2.(F), Loss Landscape smoothness is used as evidence for training stability and convergence. I have two concerns with Figure 6 and the corresponding text in 4.2.F.
i) I am not sure how the authors came to conclusion that mixup is an exception based on Fig 6.

Ii) Additionally, Loss Landscapes seem to provide an approximate visualization around local region of the final model state rather than providing stability and convergence along overall training regime. May be loss/accuracy curve along the whole training epoch might provide better evidence for convergence and training stability comparision among the mixup methods.

**Relation To Prior Work:**

Yes

**Summary And Contributions:**

The paper proposes extensive benchmarking of various mixup augmentation strategies on multiple classification datasets with varied class label granularity. The paper brings together implementations of various static and dynamic mixup strategies along with batteries-included extensible benchmarking.

Interesting insights on comparative performance, robustness against corruptions etc. are obtained based on comparison on equal-footing.

---

> ### Author Rebuttal · Authors · 2024-08-17
>
> ## Rebuttal to Reviewer aWQj
>
> ---
>
> Dear Reviewer aWQj,
>
> We would like to express our gratitude to you for raising detailed and constructive points to help us improve this work. We appreciate your recognition of our comprehensive benchmark, the valuable insights provided, the open-source code contribution, and the extensiveness of our experiments. Based on your suggestions and questions, we would address your concerns as follows:
>
> ---
>
> **(W1) Use of single class activation example with somewhat subjective conclusions.**
>
> We appreciate your critical observation regarding Figure 5 and the associated conclusions. We agree that a single example may not provide sufficient evidence for broad generalizations. We address this issue with new evaluations and clarifications:
>
> * (1) Since analyzing object localization abilities with CAM toolkits is the common practice in numerous mixup studies (stems from CutMix), we will first move Figure 5 and the associated subjective statements to the Appendix as examples of visualization tools. We will then expand our discussion and conclusions supported by quantitative analysis with the transfer learning results in Tables A11 and A12 to the main text. We also supported advanced CAM visualization with more accurate CAM variants [1, 2].
>
> * (2) We further conduct a comprehensive quantitative evaluation of localization performance across multiple objects. Following MaxBoxAccV2 [3], we evaluate the weakly supervised object localization (WSOL) task on CUB-200, where the model localizes objects of interest based on the activation maps of CAM without bounding box supervision. As shown in **Table R1** in the rebuttal PDF, we report the MaxBoxAcc (Maximal Box Accuracy) with a threshold $\delta \in$ {$0.3, 0.5, 0.7$}, which shows that dynamic mixup approaches (e.g., AutoMix and SAMix) yield a better localization of the foreground objects than the static ones (e.g., CutMix). We will provide more benchmark results based on different architectures (e.g., vision Transformers) trained on ImageNet-1K.
>
> These changes will provide a more robust and objective assessment of the localization capabilities of different Mixup methods.
>
> ---
>
> **(W2) Loss Landscape as evidence for training stability and convergence.**
>
> Thanks for highlighting these concerns. We agree that our current presentation and interpretation of the loss landscape results could be somewhat improved. We address these with two aspects.
>
> * (1) As for training stabilities, we will add a visualization of training epochs and validation performances (or losses) to support the analysis of the training process, as shown in **Figure R1** in the rebuttal PDF. In the main text, we will remove the claim about mixup methods being an exception based on Figure 6, as we agree this conclusion is not sufficiently supported. These loss/accuracy curves provide a more comprehensive view of convergence and training stability, e.g., **Figure R1 (a)(b)** shows that static mixup, dynamic mixup, and ViT-based methods have different convergence properties.
>
> * (2) As for loss landscapes, we clarify that visualization like Figure 6 provides insight into the local geometry around the final state rather than the entire training process. It can also reflect the convergence and generalization properties [4] by comparing the steepness of the margins (related to training stabilities) and the flatness of the local basin (related to the convergence and performances). Commonly speaking [5], deeper landscapes yield better performances, while wider landscapes generalize better. Figure 5 shows that AutoMix rich a deeper and wider landscape than CutMix variants, thus achieving better performances and convergence speed.
>
> We will revise our discussion in Section 4.2(F) to more accurately reflect what can be inferred from both the loss landscapes and the new training curves.
>
> ---
>
> **(W3) Additional Feedback.**
>
> We appreciate your attention to detail in identifying typos and areas for improved clarity. We will make the following changes in our final revision:
>
> - (1) Correct all mentioned typos throughout the paper.
> - (2) Move the definition of "Applicability" from the appendix to the main text, introducing it when first used.
> - (3) Simplify verbose sentences as suggested, e.g., "It is easy to select and combine datasets, mixup strategies, and schedulers."
> - (4) Remove unnecessary adjectives to improve clarity and conciseness.
>
> ### Reference
>
> [1] LayerCAM: Exploring Hierarchical Class Activation Maps for Localization. TIP, 2021.
>
> [2] Eigen-CAM: Class Activation Map using Principal Components. SN Computer Science, 2021.
>
> [3] Evaluating weakly supervised object localization methods right. CVPR, 2020.
>
> [4] Visualizing the Loss Landscape of Neural Nets. NeurIPS, 2018.
>
> [5] When Do Flat Minima Optimizers Work? NeurIPS, 2022.
>
> ---
>
> We sincerely thank you for your insightful review, which has helped us identify critical points for improvement. We hope that with these clarifications and proposed improvements, you will find our paper to be a more valuable and impactful contribution. If you have any further questions or suggestions, we would be more than happy to address them until the end of this discussion phase.
>
> Thank you once again for your efforts and constructive comments.
>
> Best regards,
>
> The Authors of Submission 630

---

> ### Author Response · Authors · 2024-08-28
> **Encouraging Discussion**
>
> Dear Reviewer aWQj,
>
> We greatly appreciate your effort and insightful feedback. In our response, we carefully answered all your questions in detail with additional evaluation and visualizations. As you may know, the discussion period this year only lasts until August 31, and we are gradually approaching this deadline. We take it seriously and would like to discuss this with you during this time if more information might be provided based on your feedback.
>
> Thanks again for your efforts in improving our manuscript! If our response could tackle your concerns, please consider updating your score in your free time. If you need any clarification, please feel free to contact us.
>
> Best regards,
>
> The Authors of Submission 630

---

### Official Review · Reviewer_r5tK · 2024-08-02
**Systematic study on the design space of Mixup augmentation is a great contribution**

**Rating:** 9
**Confidence:** 4
**Correctness:** Correct and sound
**Clarity:** Well written

**Review:**

Pro:
1) The study of the design space of Mixup augmentation is very beneficial to the visual representation learning community. Even this is agreed to be a very helpful augmentation technique, lack of clear picture of the design space can hinder further progress.
2) The study is comprehensive and solid, covering most of the existing approaches, evaluated on different datasets and model scales,  investigated the performance and cost, etc.
3) The open-sourced codebase can be helpful for the future works in the field

Con:
No major concerns, but would be nice to also have a explicit study on the performance gaps resulted from the Mixup designs. In other words, if people randomly select one of the existing design, how much worse it would be against the optimal choice.

**Strengths:**

1) The study of the design space of Mixup augmentation is very beneficial to the visual representation learning community. Even this is agreed to be a very helpful augmentation technique, lack of clear picture of the design space can hinder further progress.
2) The study is comprehensive and solid, covering most of the existing approaches, evaluated on different datasets and model scales,  investigated the performance and cost, etc.
3) The open-sourced codebase can be helpful for the future works in the field

**Additional Feedback:**

N/A

**Documentation:**

Well documented

**Ethics:**

No concern

**Limitations:**

No such limitations

**Opportunities For Improvement:**

No major concerns, but would be nice to also have a explicit study on the performance gaps resulted from the Mixup designs. In other words, if people randomly select one of the existing design, how much worse it would be against the optimal choice.

**Relation To Prior Work:**

Well discussed

**Summary And Contributions:**

The paper introduces a new benchmark to systematically study the design space of Mixup augmentation on visual representation learning/
The contributions:
1) The paper collected 18 mixup baselines and evaluated them on multiple image datasets of varying scales
2) The paper obtained some insightful observations about Mixup augmentation from the evaluations results
3) The paper developed and open-sourced a modular codebase for the mixup design.

---

> ### Author Rebuttal · Authors · 2024-08-16
>
> ## Rebuttal to Reviewer r5tK
>
> ---
>
> Dear Reviewer r5tK,
>
> Thank you for your detailed comments and the high rating on our manuscript. Your recognition of the comprehensiveness of this study and its potential benefits to the research community is truly encouraging. We would like to first express that we have been actively refining the work to make it better serve the community. We have carefully proofread the entire manuscript, correcting several typos and grammatical errors and adding some significant clarifications and discussions. We do hope to provide the community with high-quality research outputs.
>
> ---
>
> Regarding your constructive suggestion for an explicit study on the performance gaps of different mixup designs, we certainly agree this would provide additional valuable insights. We would answer your question as follows:
>
> Ideally, a Random Selection Experiment would be the most targeted approach. It would directly simulate the scenario you mentioned, where practitioners might randomly choose a Mixup technique without prior knowledge. However, we believe that the results of such experiments can be effectively inferred from our existing benchmark. On the one hand, we have provided all results in the appendix and the open-source code repository. Researchers can clearly see the **specific performance** of each Mixup method in different scenarios. We believe this may effectively reflect the performance gap between different designs. To better enhance clarity, we highlight the **best and worst-performing methods and rankings** of each method in tables.
>
> On the other hand, our benchmarking results and take-home messages are specifically designed to **eliminate this randomness**.  As mentioned by Review BRDY, OpenMixup provides clear guidance for selecting appropriate mixup strategies. Thus, practitioners in the community do not have to randomly select a Mixup method without prior information. For example, one can choose Mixup methods based on our results for general cases (where some superior dynamic methods are more recommended) or opt for Mixup+CutMix when prioritizing efficiency. It is not our intention to emphasize good or bad methods but to present fair results for practitioners to choose from. We hope this clarification addresses your concern.
>
> ---
>
> Thank you once again for your excellent comments. We believe that, after a thorough revision, this paper fits well with the NeurIPS 2024 Datasets and Benchmarks Track. We hope that this work can be seen by more researchers and practitioners, thereby benefiting the wider community. While the overall feedback has been positive, we have noticed that some points of our work may be misunderstood by a few reviewers. Thus, your rating is particularly valuable for us. We would be grateful for your further support in the discussion stage. Your active engagement would provide great support to OpenMixup, making it accessible to a broader audience.
>
> Also, we respectfully ask that you now feel more positive about our submission, and we are happy to continue the discussion and address any additional questions or concerns you may have until the end of the discussion period. Thank you!
>
> Best regards,
>
> The Authors of Submission 630

---

### Official Review · Reviewer_SK4a · 2024-08-12
**A benchmark of data augmentation mixup methods, the results show substantial complexity increases of top performing augmentation methods.**

**Rating:** 4
**Confidence:** 2

**Review:**

Covered in other sections.

**Strengths:**

The tradeoffs highlighted in sections A to G in Observations and Insights show that dynamic methods' main outperformance comes with substantial increases in computation and memory, and handcrafted methods do not generalize well across datasets and are not robust to adversarial augmentations.

**Additional Feedback:**

N/A

**Clarity:**

On line 15, the authors state that “...OpenMixup has contributed to a number of studies in the mixup community”. Please do reference them.
On line 54, start the sentence with “A benchmark”.
In Figure 2. It would be clearer if the labels of the images and the corresponding augmentation method were shared

**Correctness:**

The evaluation methods and experiment design seem appropriate. As a benchmark, alternative regularization, varying data, and compute baselines should be available on the mixing methods.

**Documentation:**

There is sufficient detail on data collection and organization, availability, and maintenance.

**Ethics:**

Some analysis and discussion on (6) Deception and harassment could be helpful, as data augmentation methods can be used to circumvent or improve AI detectors.

**Limitations:**

The authors emphasize the limitations of the work due to focusing in classification tasks only.

**Opportunities For Improvement:**

The results show dynamic methods outperform static ones, but the hyperparameters and image manipulation complexity are substantial, and it's not clear that the same performance could not be achieved through more resource consumption and data collection. The authors acknowledge this in line 49, but as a solution, they offer a benchmark. It's not clear how a benchmark helps much to address that problem. The results in Figure 4 show up to 2x memory, 5x compute, and 5x training time for the top data augmentation methods.
Some studies on how the augmentation methods compare against regularization or scaling data collection and compute would make the contributions as a benchmark more convincing.
Please elaborate on the Beta(alpha, alpha) distribution on line 16.
In Table 1. and Table 5. please share results along with ranks.
On line 297, the authors state that hand-crafted methods outperform learned ones. Please offer a hypothesis as to why. Also, on line 292, “overall capacity” is not clear. What do you mean?
Alternative regularization, varying data, and compute baselines on the mixing methods should be available as a benchmark.

**Relation To Prior Work:**

Yes.

**Summary And Contributions:**

OpenMixup is a data augmentation benchmark for visual representation learning. 18 mixup methods are evaluated on 11 datasets. Data augmentation is shown to improve generalization by smoothing the decision boundaries. The authors present experimental results on a more general task than previous work and offer an analysis of the performance-complexity tradeoffs of methods. The contributions are baseline models trained with augmentation methods, a codebase, and observations on model performance and compute and memory usage.

---

> ### Author Rebuttal · Authors · 2024-08-17
>
> ## Rebuttal to Reviewer SK4a (Part 2/2)
>
> ---
>
> **(Q9) Labeling in Figure 2:**
>
> We appreciate this suggestion for improving the clarity of Figure 2. In the revised version, we will update the figure to include clear labels for both the original images and the corresponding augmentation methods. This will enhance the reader's understanding of the visual differences between various mixup techniques.
>
> ---
>
> We sincerely thank you for your thorough review. Although there are some misunderstandings, your comments have helped us identify aspects where we can improve the clarity and comprehensiveness of our paper. We believe that, after a thorough revision, this paper fits well with the NeurIPS 2024 dataset and benchmark track, and your rating is particularly valuable to us. We do hope this work can be seen by more researchers and practitioners, thereby benefiting the wider community. Thank you once again for your time and efforts!
>
> Best regards,
>
> The Authors of Submission 630

---

> ### Author Rebuttal · Authors · 2024-08-17
>
> ## Rebuttal to Reviewer SK4a (Part 1/2)
>
> ---
>
> Dear Reviewer SK4a,
>
> We appreciate your time and effort in reviewing our OpenMixup paper. Your feedback is really detailed, and we welcome the opportunity to address your concerns and clarify key points about mixup augmentation that you may have misunderstood. We would respond to each of your questions in detail:
>
> ---
>
> **(Q1) The “solution” for performance-cost trade-offs:**
>
> Thank you for raising a good point. Indeed, this trade-off is precisely why a comprehensive benchmark like OpenMixup is crucial. As the first mixup visual representation learning benchmark, OpenMixup provides researchers and practitioners with a standardized framework to evaluate these trade-offs across various datasets and network architectures. Our goal is not to prescribe which method is universally "best", but rather to offer clear, empirically-derived observations through extensive experiments. This enables the community to make informed decisions based on specific requirements and resource constraints. The benchmark serves as a foundation for future research, allowing the community to:
>
> - (1) Understand the current landscape of mixup augmentation methods;
> - (2) Choose appropriate methods for specific deployment scenarios;
> - (3) Identify directions for improvement in method design;
> - (4) Facilitate impartial comparisons in future work;
>
> The benchmark itself is, and will never be the “solution” to the trade-off problem. It is an essential tool supporting the ongoing process of exploring “solutions” within the community. This aligns with the core contributions expected from a benchmark paper.
>
> ---
>
> **(Q2) Comparison with other regularization techniques and data collection scaling:**
>
> Thank you for this insightful suggestion. However, we would like to clarify that mixup augmentation is a distinct research area from general regularization techniques. Our benchmark is specifically designed to focus on mixup methods, which have unique properties and challenges. More importantly, there is no such benchmark to evaluate and summarize mixup methods. Expanding the scope to include all possible regularization techniques would be beyond the focus of this work and could dilute its specific contributions to the mixup field. Thus, there is no need to cover these aspects to make this work more “convincing”. However, notice that our open-source codebase is designed with extensibility, allowing for easy integration of such comparisons in future studies.
>
> We appreciate the suggestion and will consider incorporating broader comparisons in future work. However, for the current paper, maintaining a focused scope on mixup augmentation allows for a more in-depth analysis of this specific augmentation technique.
>
> ---
>
> **(Q3) Elaboration on the Beta distribution:**
>
> We appreciate your attention to detail. It seems that you might not be familiar with the field of mixup augmentation. The Beta distribution $\text{Beta}(\alpha, \alpha)$ is a fundamental concept in mixup literature [1] to sample the mixing ratio $\lambda$. It is usually regarded as a symmetric distribution on $[0, 1]$, where $\alpha$ controls the shape. This allows control over how "mixed" the samples tend to be. We recommend the Mixup paper [1] for a detailed explanation of Beta distribution.
>
> ### Reference
>
> [1] mixup: Beyond Empirical Risk Minimization, ICLR 2018.
>
> ---
>
> **(Q4) The results of the ranks in Tables 1 and 5:**
>
> We appreciate this suggestion. The rankings in these tables are derived from comprehensive experimental results, reflecting different priorities for various requirements. We have provided detailed ranking rules in the appendix to ensure transparency in our methodology.
>
> ---
>
> **(Q5) Hypothesis on why static methods surpass learned ones in applicability:**
>
> Our conclusion that hand-crafted methods generally outperform learned ones in applicability is based on our extensive benchmarking results. We argue this may come from their simplicity and lower computational requirements. Static methods don't require additional optimization steps or complex architectures, making them easier to implement and faster to execute.
>
> ---
>
> **(Q6) Clarification on "overall capacity":**
>
> Please refer to (Q4). This term refers to a holistic evaluation of a method's utility, considering both its performance improvements and practical applicability. We have provided detailed ranking rules in the appendix, which explain how we quantify and combine these factors.
>
> ---
>
> **(Q7) Alternative regularization, varying data, and compute baselines:**
>
> We appreciate this suggestion but would like to reiterate that our benchmark focuses on mixup augmentation methods. While broader comparisons with other regularization techniques could provide interesting insights, they fall outside the scope of this particular study.
>
> Our goal is to provide the first comprehensive benchmark within the mixup domain, which has its own unique characteristics and challenges. Our open-source codebase does allow for future extensions to include such comparisons, which could be valuable for future studies. Please refer to (Q2) for more details.
>
> ---
>
> **(Q8) References for studies that have used OpenMixup:**
>
> We understand your interest in concrete examples of OpenMixup's impact. In contrast, it's unconventional to include citations in the abstract. It is effortless to find out if we overclaim or not. Additionally, the project's GitHub and Google Scholar page provides real-time information on its adoption and impact on the community. But for clarification, we provide several references here to substantiate our claim. Thank you for your kind suggestions and reasonable concerns.
>
> ### Reference
>
> [1] SMMix: Self-motivated Image Mixing for Vision Transformers. ICCV 2023.
>
> [2] Harnessing Hard Mixed Samples with Decoupled Regularizer. NeurIPS 2023.
>
> [3] Adversarial AutoMixup. ICLR 2024.
>
> [4] SUMix: Mixup with Semantic and Uncertain Information. ECCV 2024.

---

> ### Author Response · Authors · 2024-08-28
> **Encouraging Discussion**
>
> Dear Reviewer SK4a,
>
> We greatly appreciate your effort and valuable feedback. In our response, we carefully illustrated and answered all your questions in detail and improved our manuscript. As you may know, the discussion period this year only lasts until August 31, and we are gradually approaching this deadline. We take it seriously and would like to discuss this with you during this time if more information could be provided based on your feedback.
>
> Thanks again for your efforts! If our response tackles your concerns, please consider updating your score in your free time. If you need any clarification, please feel free to contact us in the last few days.
>
> Best regards,
>
> The Authors of Submission 630

---

### Author Rebuttal · Authors · 2024-08-17

## General Response by Authors

---

Dear Reviewers,

Greetings!

We sincerely thank all **six reviewers** for the insightful feedback on our submission. Your comments have been exceptionally helpful in improving our work. Based on your feedback, we have individually responded to each reviewer's comments regarding the **value and benefits for both the research and engineering communities**. Meanwhile, we addressed problems regarding the experiment setup and evaluation results, as well as presentation clarity mentioned by some reviewers in the revision. With the help of the reviewers, our revision more accurately illustrates the problems we addressed, the contributions we made, and the empirical observations we obtained.

---

To further elaborate on the significance of this work and outline the improvements made in our revision (which cannot be uploaded this year), we offer the following clarifications:

**(1) Technical Contribution.** OpenMixup represents **the first** comprehensive mixup visual classification benchmark. It enables researchers to **fairly compare** their models against a broad spectrum of mixup baselines, providing **a clear measurement** of the methods’ effectiveness. As a fundamental building block of today's computer vision system, there has been no such comprehensive benchmarking study for mixup augmentation, which has been a notable gap in the community. As researchers, we believe we can all agree that an area will struggle to move forward without a comprehensive benchmark. More importantly, such a first-of-its benchmarking study is both **time-intensive** and **resource-demanding**, but we still carried it out from scratch for the community.

**(2) Open-source Codebase and Reproducibility.** We have developed a standardized model design and training codebase platform for mixup-based visual representation learning, ensuring **fair and reproducible comparisons** across various datasets. The overall framework is streamlined and modular, which can enhance the **accessibility and customizability** of downstream deployment and applications. As suggested by *Reviewer TiUv,* we have further verified the reproduction reliability.

**(3) Multiple Evaluation Metrics.** OpenMixup incorporates standard evaluation pipelines of image classification, robustness, calibration, and transfer learning with computational metrics. These multifaceted results show consistent conclusions with the accuracy performance while introducing diverse analytical perspectives for practical usage. In response to *Reviewer aWQj*'s suggestion, we further provided a quantitative evaluation of the localization capabilities of different mixup methods.

**(4) Analysis and Observations.** We conducted empirical analyses across various scenarios. Interesting **observations and insights** are obtained (refer to **Sec.4**) with diverse analysis toolkits (e.g., accuracy radar plots, trade-off plots, loss landscape, CAM visualization, etc.) provided in our codebase, enabling researchers to identify what specific properties contribute to the success of different methods. As suggested by *Reviewer aWQj,* we have added plots of training epochs *vs.* accuracy to analyze training stability. Consequently, we summarize our findings on effective mixup algorithms from six key aspects in Sec. 4.2, offering practical guidance for researchers and practitioners.

---

Overall, we appreciate all your excellent comments. We hope that you can reconsider our submission with the rebuttal feedback. If you are satisfied with our work and response, please consider updating your ratings. We do hope OpenMixup can reach more researchers and practitioners, thus benefiting a broader community. Should you require any further clarification, please feel free to contact us. Thank you once again for your time and effort!

Best regards,

The Authors of Submission 630

---

### Author Response · Authors · 2024-08-23
**Encouraging Discussion and Feedback**

Dear Reviewers,

We greatly appreciate your valuable feedback on the manuscript and your recognition of the contributions of our proposed benchmark. In our response, we carefully answered all questions and tackled concerns in detail. Additional experiment results and revised paragraphs are also provided, as you requested. We further clarified several unclear statements in the paper to demonstrate our contributions clearly. We hope your concerns have been addressed.

As you may know, with increasing amounts of submissions, the discussion period this year only lasts until August 31, and we are gradually approaching this deadline. We take it seriously and would like to discuss this with you during this time. If you are satisfied with our response, please consider updating your score. If you need any clarification, please feel free to contact us. We would happily provide more information based on your comments or further questions.

Warm regards,

The Authors of Submission 630

---

### Decision · Program_Chairs · 2024-09-26

**Decision:**

Reject

**Comment:**

This paper received mixed scores and was ultimately considered borderline. While some positive reviews from the reviewers are respected, this AC believes crucial points still need to be addressed. Therefore, this AC conducted an independent review to ensure the right decision.

The paper includes extensive experiments with various MixUp-based methods, and the AC acknowledges the value of the results. However, the AC also identifies several key concerns:

1) Insufficient evaluation: The paper lacks metrics beyond performance to analyze the methods, limiting the depth of the evaluation.
2) Insufficient analysis: Beyond the loss and cam visualizations, more comprehensive analysis should be required to provide deeper insights. Offering more insightful evaluation metrics or analyses would give refreshing takeaways.
3) On paper's scope: The paper focuses solely on image classification (with transfer learning being an outcome). While crucial, expanding the method's applicability to other tasks would be appealing; additionally, the robustness metrics provided are mostly tied to image classification performance, which limits the ability to fully reveal the nature of the variants.
4) On promoting dynamic methods: The rationale behind promoting dynamic methods like SAMix or AutoMix, which demand high training costs, is unclear. As noted by Reviewers SK4a and BRDY, these methods appear on the Pareto curve in terms of cost-effectiveness and may not be the best option for broader applicability.

Overall, despite the significant contribution of extensive experiments with MixUp variants, these concerns prevent this AC from recommending acceptance in this round. Furthermore, this AC believes that more scientifically strong papers that offer more takeaways should be prioritized.